# DDCoT: Duty-Distinct Chain-of-Thought Prompting for Multimodal Reasoning in Language Models

Ge Zheng[1][*]  Bin Yang[1][*]  Jiajin Tang[1][*]  Hong-Yu Zhou[2]  Sibei Yang[1][†]

[1]ShanghaiTech University    [2]The University of Hong Kong

Project Page: https://toneyaya.github.io/ddcot/

## Abstract

A long-standing goal of AI systems is to perform complex multimodal reasoning like humans. Recently, large language models (LLMs) have made remarkable strides in such multi-step reasoning on the language modality solely by leveraging the chain of thought (CoT) to mimic human thinking. However, the transfer of these advancements to multimodal contexts introduces heightened challenges, including but not limited to the impractical need for labor-intensive annotation and the limitations in terms of flexibility, generalizability, and explainability. To evoke CoT reasoning in multimodality, this work first conducts an in-depth analysis of these challenges posed by multimodality and presents two key insights: *"keeping critical thinking"* and *"letting everyone do their jobs"* in multimodal CoT reasoning. Furthermore, this study proposes a novel DDCoT prompting that maintains a critical attitude through negative-space prompting and incorporates multimodality into reasoning by first dividing the reasoning responsibility of LLMs into reasoning and recognition and then integrating the visual recognition capability of visual models into the joint reasoning process. The rationales generated by DDCoT not only improve the reasoning abilities of both large and small language models in zero-shot prompting and fine-tuning learning, significantly outperforming state-of-the-art methods but also exhibit impressive generalizability and explainability.

## 1 Introduction

One of the fundamental aspirations of AI systems is to address complex tasks with reliability and efficiency that are aligned with human capabilities [31, 48, 58, 11, 5, 52, 53, 49, 50]. In tackling such complex tasks, humans rely on multi-step reasoning that integrates information from various modalities. Recently, language models (LMs) [3, 37, 55, 8, 7] have shown remarkable progress in a range of multi-step reasoning tasks by prompting [60, 25, 70, 15, 54] or fine-tuning [35, 13, 59, 27] with the chain of thought (CoT) that mimics the human reasoning process.

However, most studies on CoT reasoning have focused solely on the language modality [44, 10, 32, 72, 22, 5, 68, 66], with minimal attention to multimodal contexts. UnifiedQA [31] and MM-CoT [71] are the pioneering works of eliciting CoT reasoning in vision-and-language multimodality. They employ multimodal inputs, as illustrated in Figure 1(a), necessitating training for the generation of the intermediate reasoning steps (*i.e.*, rationales) either as an explanation accompanied by an answer or as reasoning prior to inferring the answer. Despite the benefits of these multimodal CoT reasoning methods in enhancing the ability of LMs to answer multimodal science questions, several significant challenges have impeded their deployment: (1) **Labor-intensive annotation**: manual annotation for rationales is time-consuming, costly, and challenging to ensure consistency and completeness in

---

[*]Equal contribution.

[†]Corresponding author. yangsb@shanghaitech.edu.cn

37th Conference on Neural Information Processing Systems (NeurIPS 2023).

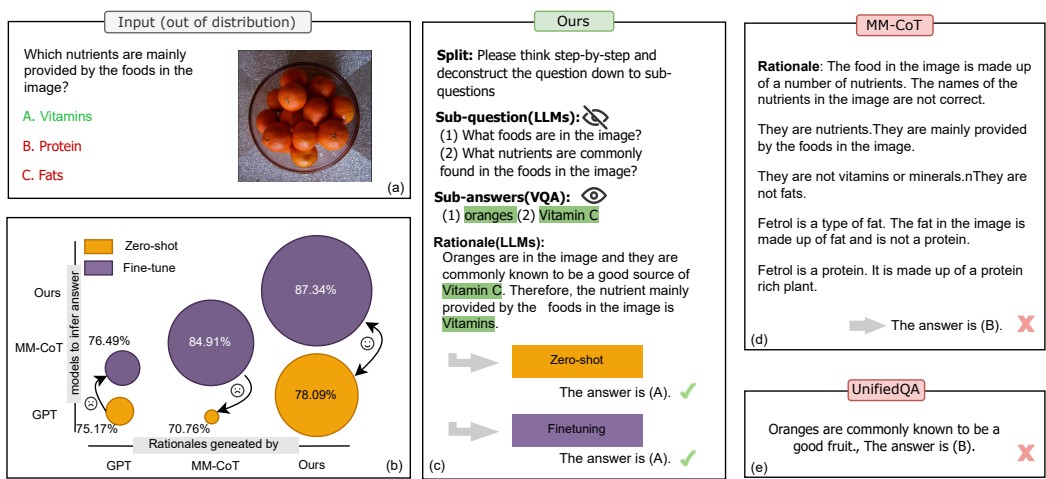

Figure 1: Comparison of existing multimodal CoT methods and our DDCoT on (a), (c), (d) and (e) an out-of-distribution example to illustrate the generalizability and (b) performance of zero-shot and fine-tuning learning, demonstrating that we are the first to generate general multimodal rationales.

multi-step CoT reasoning, thereby restricting the utilization of these approaches. (2) **Flexibility**: the effectiveness of multimodal rationales generated by existing methods [31, 71] is limited to either zero-shot prompting or fine-tuning learning with LMs, as demonstrated in Figure 1(b). Our observations indicate that the rationales generated by MM-CoT fail to provide any benefits for zero-shot prompting, while the rationales generated solely by GPT-3 do not enhance the reasoning ability of the fine-tuning model. (3) **Generalizability**: the current multimodal CoT reasoning approaches [31, 71] exhibit limited generalizability to problems that require novel and unseen reasoning paths. As shown in Figure 1(d) and (e), basic science questions that are out-of-distribution pose a challenge for these models. Moreover, models trained on a specific subset, which includes arbitrary two subjects of questions in natural science, social science, and language science, perform worse when applied to another subject of questions (see Section 4.1). (4) **Explainability**: the objective of multimodal CoT extends beyond inferring answers to include providing explanations, yet the interpretability of current generated rationales still requires further improvement.

This study aims to explore overcoming the aforementioned challenges and develop a *zero-shot*, *generalizable* and *explainable* approach for generating rationales to enhance the reasoning abilities of LMs in *both zero-shot and fine-tuning learning*. To achieve this, we first probe the the incorporation of rationales into multimodal reasoning and determine a two-step rationale generation and utilization process. Then we explore (1) the role of rationales in the utilization phase, where they serve as authentic knowledge providers, and (2) the challenges of intensified hallucinations in rationale generation phase. Based on these factors, we formulate the following approach.

To generate general multimodal rationales, we propose a novel multimodal CoT prompting approach called Duty-Distinct Chain-of-Thought Prompting (DDCoT), which generate multimodal rationales using language-only LLMs [3, 37], considering the explored factors above. For the first factor, our key insight is *"critical thinking – keeping a healthy dose of skepticism"*: explicitly indicating the uncertainty in the rationale generation process is critical since it helps improve the correctness of the generated rationales, which serve as key inputs guiding LMs in the thinking process of inferring answers, particularly in the zero-shot setting. For the second factor, our key insight is to *"let everyone do their jobs – divide labor and work together"*: it is necessary to prompt LLMs to explicitly identify the reasoning and recognition responsibility of LLMs and off-the-shelf visual models, in order to overcome the language hallucination issue that arises when directly generating rationales from interleaved multimodal inputs. Based on this, we propose a sequential process of negative-space prompting, visual recognition, and joint reasoning to perform interleaved reasoning and recognition. More detailed analysis of the observed factors and motivation behind the two insights are elaborated in Section 3.1 and Section 3.2.

To utilize the generated rationales to facilitate multimodal question answering of LMs, we leverage them as inputs to explicitly guide the LMs' chain of thought and the attention to multimodal inputs in both zero-shot prompting and fine-tuning learning. We combine generated rationales with problem

statements as inputs to the LLMs for zero-shot learning. For fine-tuning learning, we propose deep-layer prompting (DLP) and rational-compressed visual embedding (RCVE) to utilize the rationales better to filter, encode, and jointly infer over interleaved multimodal inputs.

To sum up, our contributions are three-fold: (1) This work is the first to study the zero-shot multimodal rationale generation. We deeply analyze the challenges and insights in multimodal CoT for the rationale generation: rationale sensitive in zero-shot prompting, the knowledge-required in fine-tuning due to catastrophic forgetting, and hallucinations intensified due to interleaved multimodal inputs. We hope these insights could benefit future research work. (2) We propose a novel DDCoT prompting to maintain a critical attitude and identify reasoning and recognition responsibilities through the combined effect of negative-space design and deconstruction. The resulting rationales can directly be part of multimodal inputs to improve the reasoning abilities of LMs in both zero-shot prompting and fine-tuning learning. (3) With the rationales, our methods consistently outperform the state-of-the-art LMs, improving both GPT-3 and UnifiedQA by +2.53% and +8.23% respectively on questions that have image context, while exhibiting impressive generalizability and explainability.

## 2  Related Work

**CoT Reasoning of LLMs.** LLMs have been demonstrated successful in natural language processing. Recently, zero-shot [25] and few-shot [60, 44] multi-step thinking prompts have been found to significantly improve LLMs' reasoning ability, thus such chain-of-thought (CoT) methods have attracted growing interest. Some works are interested in example selection in terms of similarity [44, 32], diversity [70], and complexity [10]. Other methods optimize the reasoning pipeline, exploring to introduce programming approach [5], explicit decomposition of problems [72, 22] or calibration coordination across multiple rationales [59, 27, 10]. Inspired by these works, we focus on extending CoT reasoning to multimodality, whilst tackling inherent complications that emerge from therefrom.

**Transferring Specialized Reasoning Skills to Small Models.** In addition to research on CoT reasoning in LLMs, some studies conduct CoT on models with a smaller number of parameters. The works [35, 13] distill the CoT capabilities of LLMs into smaller models, facilitating the performance of smaller models in specific tasks. Unlike them to generate rationales in inference, our work utilizes the generated rationales via CoT reasoning from LLMs to explicitly guide the understanding of the image for multimodal reasoning.

**Cross-modal CoT Reasoning.** The pioneering work [31] for multimodal CoT proposes ScienceQA, a dataset consisting of scientific questions involving multimodality with annotated rationales. They perform zero-shot prompting on GPT-3 and fine-tuning learning on UnifiedQA [20] to generate rationales and answers simultaneously.

The following work MM-CoT [71] devises a two-stage framework in which the model initially learns to generate rationales based on the ground-truth annotations, and then utilizes all available information to generate the final answer. However, their generated rationales can only exclusively benefit either zero-shot or fine-tuning learning.

**Integrate Visual Modality to Language Models.** With the demonstrated capability of LLMs to incorporate open-world commonsense knowledge [40, 43], an increasing number of studies are focused on making visual modality available to well-established LLMs for solving complex visual and multimodal problems. Some approaches [16, 26, 67, 28, 69, 6, 12, 63] incorporate additional training data to align image features with linguistic space. Others [33, 62, 47] take advantage of the scheduling ability of LLMs to dynamically integrate off-the-shelf vision models, extracting image information into text form explicitly. Unlike them directly integrating multimodal inputs once, we explicitly identify the uncertainty in rationale generation and complement visual information step by step.

## 3  Method

Our work focus on how to best put visual information in the text to generate generalizable and explainable rationales. In this section, we first introduce the concepts and motivation in exploration (Sec 3.1), and then present our concrete method designs for rationale generation (Sec 3.2) and utilization (Sec 3.3).

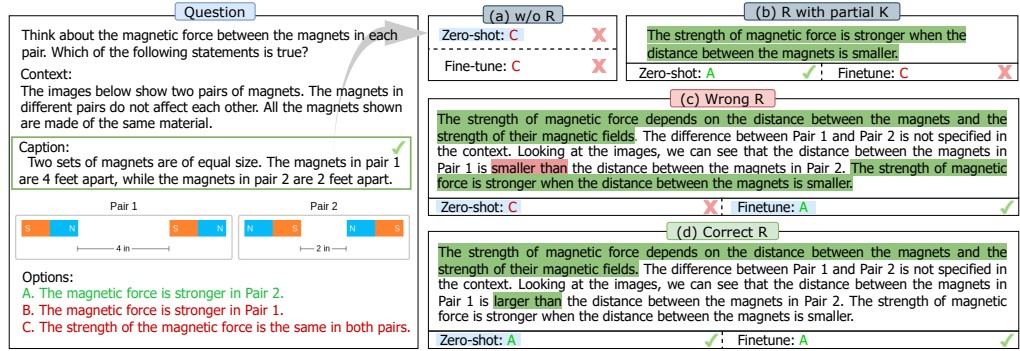

Figure 2: An example shows the importance of input rationales in multimodal reasoning as well as the disparate roles of rationales in zero-shot and fine-tuning scenarios.

## 3.1 Motivation: Leverage Rationales for Multimodal Reasoning

The success of rationales in unimodal reasoning [25, 60, 44] motivates us to leverage rationales to enhance both reasoning capabilities and interpretability in multimodal reasoning. In this section, we first investigate the incorporation of rationales into multimodal reasoning in Section 3.1.1, and then examine the roles of rationales in enhancing multimodal reasoning in Sections 3.1.2, while also illuminating the challenges to generate general rationales with LLMs in Section 3.1.3.

### 3.1.1 Two-step Reasoning Process: Multimodal Rationales Generation and Utilization

Following [31, 71, 69], we first prompt LLMs to generate answers accompanied by rationales and observe that it cannot improve multimodal reasoning capabilities, *i.e.*, 74.04% accuracy of generating only answers and 75.17% accuracy of generating both rationales and answers [31].

Moreover, as Figure 2(a) illustrates, despite sufficient image information provided in the form of image caption, the LLM, GPT-3 [3] in this case, fails to jointly reason the question and the caption to infer answer. The possible reason is that LLMs have difficulty understanding dense image information.

Motivated by that people usually extract key information from images in the context of questions, rather than indiscriminately focusing on all dense information, *we explore to provide the rationale as a structured logical chain to explicitly guide the understanding of the image.* As shown in Figure 2(a) and (d), incorporating the rationale as input facilitates the model's multimodal reasoning capabilities. Therefore, we employ a two-step reasoning process: multimodal rationale generation, and their subsequent utilization.

### 3.1.2 Roles of Rationales Differ in Zero-shot and Fine-tuning Learning

In further exploration, we find that the effect of rationale varies in zero-shot and fine-tuning models.

**Rationale-sensitive reasoning for zero-shot prompting.** Upon conducting zero-shot prompting on LLMs, such as ChatGPT [37], we find that the model tends to reason in line with the input rationales. For example, without referring to additional rationales, the common sense knowledge encoded in ChatGPT allows it to answer the basic biological question "which nutrients are mainly provided by the foods in the image", as shown in Figure 3(c). However, the assertion "oranges are known to be a good source of fats" in misleading rationale results in a failure of ChatGPT to answer the same question (see Figure 3(a)). Given language models' tendency to reason primarily based on input rationales, the accuracy of rationales becomes crucial for zero-shot prompting on LLMs.

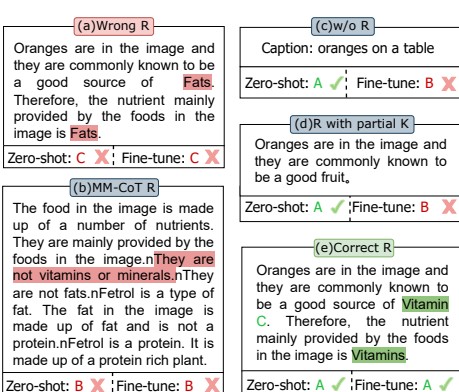

Figure 3: An example for disparate roles of rationales in zero-shot and fine-tuning scenarios, with question depicted in Figure 1.

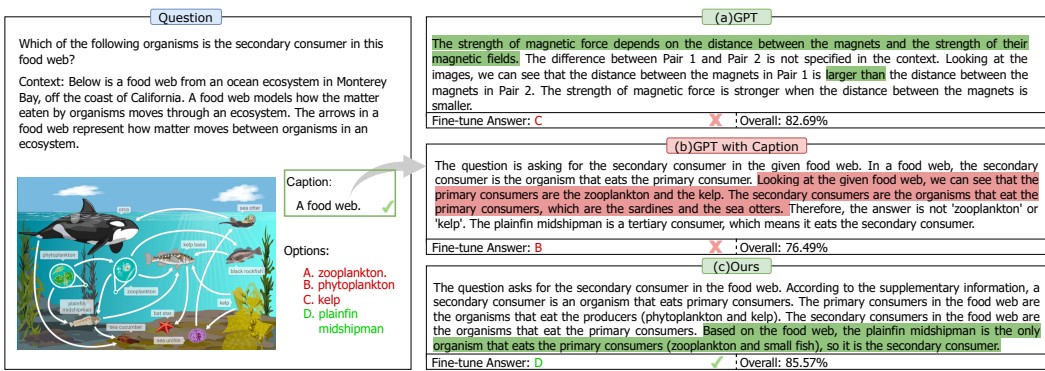

Figure 4: An example of the hallucination challenge which is exacerbated by providing interleaved multimodal information at once in generating rationales with LLMs.

**Knowledge-required reasoning for fine-tuning.** In contrast to zero-shot prompting learning, the fine-tuning models not only require the correctness of rationales but also exhibit a strong reliance on comprehensive prior knowledge embedded in rationales. As shown in Figure 3(d) and 2(b), lacking common sense knowledge, such as the nutrient mainly provided by oranges and factors affecting magnetic force, the fine-tuning model fails to answer these two questions. Compared to zero-shot prompting, the fine-tuning model showcases enhanced error tolerance (as shown in Figure 2(c)), while concurrently showing increased susceptibility to knowledge deficiencies. This stems from the catastrophic forgetting during fine-tuning in LMs [14, 4, 65].

### 3.1.3 Challenge Lies in LLMs with Multimodality: Hallucinations Intensified

To generate general rationales simultaneously fulfill the above two roles for assisting language models in understanding multimodal information, we first analyze the state-of-the-art method [71], which is trained to generate rationales relying on manual annotations. However, it generates rationales without considering their correctness, thereby leading to the risk of misleading language models. In addition, based on training with manually annotated rationales, it lacks the ability to generate rationales that encompass sufficient relevant knowledge for out-of-distribution questions, leading to a significant performance drop (see Figure 3(b) and the left table in Table 2). Unlike MM-CoT [71], we resort to LLMs to generate rationales with zero-shot prompting, leveraging their intrinsic capacity for generalization. We further explore the strategies to generate rationales.

**Uni-modal rationales have limited effect.** We start by directly prompting *"let's think step by step"* without any image information, resulting in visually irrelevant rationale (see Figure 4(a)) and poor performance (see row "w/ naive R" in the right table in Table 2).

**Interleaved information exacerbates hallucinations.** To generate multimodal rationales involving image information, the challenge lies in mitigating the language hallucinations [24, 36, 17, 2], which is exacerbated by providing interleaved multimodal information at once. A naive attempt is to extract the image caption and combine it with the question as a joint prompt to generate rationales. Despite the integration of image information, the generated rationales remain suboptimal for image-related questions, which is not aligned with our initial expectations. Upon analyzing various cases, we observe that interleaved information predisposes LLMs towards hallucination, generating fabricated visual information. As illustrated in Figure 4, we extracted the caption "a food web" from the image, which lacks the necessary information. Consequently, the language model resorts to imagining image-related information such as "the primary consumers are the zooplankton and the kelp" (see Figure 4(b)), leading to difficulty in distinguishing reliable knowledge from the language hallucinations.

## 3.2 Zero-shot DDCoT Prompting for Multimodal Rationale Generation

We generate general multimodal rationales based on the following key insights, which are derived from Section 3.1:

(1) Utilize two-step reasoning process, considering that LLM has difficulty jointly reasoning multi-modal information directly (Section 3.1.1).

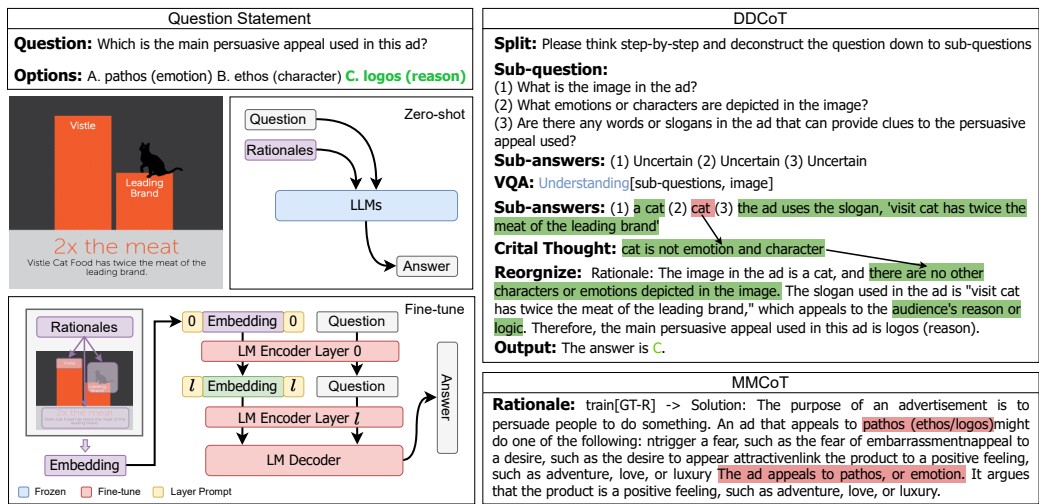

Figure 5: An overview of our DDCoT and its utilization to improve the multimodal reasoning of LMs. Note that although errors encounter in the second sub-problem during visual recognition, the language model rectifies this error in the joint reasoning step with critical thought.

(2) Generate rationales that meet the requirements of both zero-shot and fine-tuning learning, filled with knowledge and requiring fidelity (Section 3.1.2).

(3) Alleviate the intensified hallucinations from interleaved information (Section 3.1.3).

Accordingly, we propose Duty-Distinct Chain-of-Thought Prompting (DDCoT), which encompasses three steps: (1) We utilize LLMs' intrinsic knowledge to generate multimodal rationales. (2) We explicitly cue the LLMs to differentiate the responsibilities of reasoning and recognition step by step. (3) We explicitly mark the negative space for uncertain parts, emphasizing critical thinking in rationale generation. Our DDCoT jointly exploits the reasoning ability in LLMs and the image understanding capability of visual question-answering models for general multimodal rationale generation.

**Breaking Reasoning Down to Recognition Steps with Negative-Space Prompting.**

First, with the given question, context, and options, we prompt the LLM [37] to deconstruct the input question into a sequence of basic sub-questions, breaking the complex reasoning chain into simple steps, as shown in Figure 5. Diverging prior works in NLP community [9, 19], we introduce the following prompting designs: (1) We utilize a single-stage deconstruction to simplify the problem-solving process. Specifically, we employ the instruction *"please think step-by-step and deconstruct the question down to necessary sub-questions"* to obtain the sub-question sequence at once. (2) Then we explicitly prompt the LLMs to determine whether each sub-question can be answered without visual information. In cases where sub-questions involving visual recognition are unanswerable, the LLMs are instructed to answer "uncertainty" as a negative space. We provide the model with the following prompt: *"Assume that you do not have any information about the picture, try to answer the sub-question and formulate the corresponding sub-answer as 'Uncertain' if the sub-question cannot be determined"*. By adopting a critical stance towards sub-questions and introducing the explicit assumption of invisibility, we successfully mitigate hallucination in LLMs when dealing with sub-questions involving images, reducing factual errors.

**Visual Recognition to Obtain Visual Complements.** However, the obstruction caused by negative space prevents LLMs from directly activating the chain-of-thought reasoning. In order to fill the negative space, we leverage the image understanding capability of the off-the-shelf models to acquire visual information as visual complements. Specifically, we employ the visual question answering (VQA) model [26] to individually answer the sub-questions with negative space, which correspond to simple visual recognition problems. Note that our approach can solely leverage the basic visual recognition capability of the VQA model and robust to its possible inference errors thanks to the following joint reasoning integration.

**Integrate to Joint Reasoning.** With a sequence of complete sub-answers including visual recognition results, we resort to LLMs again to integrate the information and engage in reasoning processes

to derive the rationales. Incorporating our obtained sub-questions and corresponding sub-answers as supplementary information, we prompt LLMs *"think step by step"* to perform joint reasoning with linguistic and visual information and generate multimodal rationales. Furthermore, we prompt explicitly with *"note that the supplementary information given may not always be valid"* and *"select valid information to form the rationale"* to encourage a critical attitude towards the supplementary information and thoughtful analysis. By explicitly highlighting uncertainty, LLMs are able to pay attention to the origin question information and intrinsic knowledge, effectively filtering and even rectifying supplementary knowledge to derive more reasonable answers, as shown in Figure 5.

### 3.3 Utilization of Multimodal Rationales in General Scenarios

In this section, we introduce the utilization of our rationales to achieve multimodal reasoning, both for zero-shot and fine-tuning learning. For zero-shot learning, we consider prompting the model with rationales, including established knowledge and a reasoning chain of thought. Additionally, for fine-tuning, our proposed rationales, in combination with our proposed deep-layer prompting and rationale-compressed visual embedding, improve deep multimodal understanding and reasoning.

**Utilization for Zero-shot Prompting.** As shown in Figure 5, for LLMs like ChatGPT, we utilize zero-shot prompts that combine rationales generated by Section 3.2 with our problem statements as inputs to the model. Our rationales generated by explicitly questioning the uncertainty with improving reliability can facilitate LLMs to make more precise assessments and exhibit fewer hallucinations.

**Utilization for Fine-tuning Learning.** The framework of the fine-tuning process is shown in Figure 5. During fine-tuning, we introduce learnable prompts in both shallow and deep layers to align cross-modal information and facilitate multimodal reasoning. In addition, instead of directly inputting the entire image into the LM, we use the multimodal rationales to guide the filtering of key image features as visual input embeddings for the LM.

- **Deep-Layer Prompting (DLP)** is designed to assist in the alignment and joint reasoning of multimodal inputs at multiple levels. It employs not only learnable prompts to facilitate the alignment of visual and linguistic semantics at a shallow level [74, 73, 51, 34] but also utilizes explicit rationales to jointly encode multimodality by learning different prompts for each encoder layer [18, 21, 30, 29]. Specifically, we randomly initialize learnable prompts $P \in \mathbb{R}^{L \times N_p \times C}$ for $L$ encoder layers. For the $l$-th layer, we load the prompt $P_l \in \mathbb{R}^{N_p \times C}$ at the begging and end of the visual input embeddings.

- **Rational-Compressed Visual Embedding (RCVE).** Instead of inputting visual features into the LM directly, we compress visual input embeddings according to our multimodal rationales. Specifically, we leverage the rationales to jointly comprehend the textual and visual contexts, serving as the prior knowledge to filtering visual features. Given the text embeddings $T \in \mathbb{R}^{N_t \times C}$ that contains context and rationale, the global visual input $V_g \in \mathbb{R}^C$, and the local visual inputs $V_g \in \mathbb{R}^{N_v \times C}$, we first update the global visual feature based its similarity to text embeddings as follows,

$$V_t = \text{Attention}(V_g, T), \tag{1}$$

where $V_t \in \mathbb{R}^C$ is the updated visual feature and $\text{Attention}(\cdot, \cdot)$ is the standard multi-head cross-attention module [56]. Next, instead of directly using the updated visual feature $V_t$ to capture relevant local visual inputs $V_l$, we introduce low-rank intermediate vectors as crucial mediators for filtering local inputs as follows:

$$V_r = \text{reshape}(\text{MLP}(V_t)), V = \text{Attention}(V_r, V_l), \tag{2}$$

where $\text{MLP}(\cdot)$ denotes a three-layer linear layer with activation function, $V_r \in \mathbb{R}^{N_r \times C_r}$ represents $N_r$ low-rank vectors, and $V$ is the final visual embeddings to input to the LM's encoder.

## 4 Experiment

**ScienceQA benchmark [31]** is the first multimodal science question-answer dataset comprising 21,000 questions with multiple choices and images. Following previous works [71, 31], we divide ScienceQA into training, validation, and test sets, which contain 12,726, 4,241, and 4,241 examples, respectively. Additionally, the questions in ScienceQA can be categorized into three domains: natural science (NAT), social science (SOC), and language science (LAN).

**Implementation.** We conduct zero-shot experiments on LLMs including ChatGPT [37] and GPT-3 [3]. Since LLMs are unable to accept images as visual input, we convert images into captions

| Model | Report | Size↓ | GT-R | NAT | SOC | LAN | TXT | IMG | NO | G1-6 | G7-12 | Avg |
|---|---|---|---|---|---|---|---|---|---|---|---|---|
| *Heuristic baselines* | | | | | | | | | | | | |
| Random [31] | NeurIPS[22] | - | - | 40.28 | 46.13 | 29.25 | 47.45 | 40.08 | 33.66 | 39.35 | 40.67 | 39.83 |
| Human [31] | NeurIPS[22] | - | - | 90.23 | 84.97 | 87.48 | 89.60 | 87.50 | 88.10 | 91.59 | 82.42 | 88.40 |
| *Few-shot models* | | | | | | | | | | | | |
| GPT-3(CoT) [31] | NeurIPS[22] | 175B | ✓ | 75.44 | 70.87 | 78.09 | 74.68 | 67.43 | 79.93 | 78.23 | 69.68 | 75.17 |
| ChatGPT(CoT) [33] | arXiv[23] | 175B | ✓ | 78.82 | 70.98 | 83.18 | 77.37 | 67.92 | 86.13 | 80.72 | 74.03 | 78.31 |
| Chameleon(ChatGPT) [33] | arXiv[23] | 175B | ✓ | 81.62 | 70.64 | 84.00 | 79.77 | 70.80 | 86.62 | 81.86 | 76.53 | 79.93 |
| *Zero-shot models* | | | | | | | | | | | | |
| Ours(GPT-3) | | 175B | | 78.60 | 73.90 | 80.45 | 77.27 | 69.96 | 82.93 | 80.65 | 73.50 | 78.09 |
| Ours(ChatGPT) | | 175B | | 80.15 | 76.72 | 82.82 | 78.89 | 72.53 | 85.02 | 82.86 | 75.21 | 80.15 |
| Fine-tuning models ▽ | | | | | | | | | | | | |
| *LLaMA based Fine-tuning models* | | | | | | | | | | | | |
| LMAdapter [69] | arXiv[23] | 7B | | 84.37 | 88.30 | 84.36 | 83.72 | 80.32 | 86.90 | 85.83 | 84.05 | 85.19 |
| *T5 based Fine-tuning models* | | | | | | | | | | | | |
| UnifiedQA [31] | NeurIPS[22] | 223M | ✓ | 71.00 | 76.04 | 78.91 | 66.42 | 66.53 | 81.81 | 77.06 | 68.82 | 74.11 |
| MMCoT [71] | arXiv[23] | 223M | ✓ | 87.52 | 77.17 | 85.82 | 87.88 | 82.90 | 86.83 | 84.65 | 85.37 | 84.91 |
| UnifiedQA [31] | NeurIPS[22] | 223M | | 68.16 | 69.18 | 74.91 | 63.78 | 61.38 | 77.84 | 72.98 | 65.00 | 70.12 |
| MMCoT† [71] | arXiv[23] | 223M | | 82.51 | 77.73 | 82.82 | 81.09 | 75.11 | 85.30 | 81.64 | 80.95 | 81.40 |
| Ours | | 223M | | **88.72** | **86.84** | **84.91** | **87.59** | **83.34** | **88.08** | **88.58** | **85.10** | **87.34** |
| | | | | +6.21 | +9.11 | +2.09 | +6.50 | +8.23 | +2.78 | +6.94 | +4.15 | +5.94 |

Table 1: Main results (%). Size = backbone model size. GT-**R** means models are trained with ground truth rationales. Question classes: NAT = natural science, SOC = social science, LAN = language science, TXT = text context, IMG = image context, NO = no context, G1-6 = grades 1-6, G7-12 = grades 7-12. † denotes implementation by removing ground truth rationales when fine-tuning.

via BLIP-2 [26] as extra input to LLMs. Furthermore, following previous works [71, 31], we adopt UnifiedQA [20] as our base model for fine-tuning. We use CLIP ViT-L/14 [41] as the visual encoder to extract the global visual input $V_g$ and the local visual input $V_l$. The hyperparamters $N_p$, $N_r$ and $C_r$ are 3, 16 and 4, respectively. We train our model for 30 epochs with a learning rate of 1e-4 and batch size of 16. All experiments are implemented by PyTorch [39] and HuggingFace [61] and conducted on NVIDIA Tesla A40 GPUs. We employ accuracy as our evaluation metric and furnish comprehensive results across each domain.

## 4.1 Comparison with State-of-the-Arts in Zero-shot Prompting and Fine-tuning Learning

Table 1 shows the comparison of our DDCoT with the state-of-the-art models [31, 33, 71] on zero-shot and fine-tuning benchmarks. Our approach consistently achieves superior performance compared to previous methods. Note that we further report the results of the recent works [33, 71, 69] which have not been published but have been preprinted in the research community.

**Zero-shot Prompting.** Regarding zero-shot prompting, our zero-shot approach outperforms the published state-of-the-art few-shot method [31] by 2.92% on GPT-3 and 1.84% on ChatGPT, respectively. Compared with the concurrent work Chameleon [33], which incorporates a vast array of external tools, our method maintains comparable performance, even attaining a 1.73% enhancement on the IMG split. These suggest that our DDCoT, integrated with negative-space prompting, effectively curtails factual errors, guaranteeing the accuracy of multimodal reasoning in LLMs. Furthermore, we observe that the performance of GPT-3 (67.43%) and ChatGPT (67.92%) in a few-shot manner is similar, and the amplification of our technique in the IMG branch escalates as the performance of the language models strengthens (2.53% for GPT-3 and 4.61% for ChatGPT). The possible reason is language models rarely acquire an innate understanding of dense image information, while explicit rationales guidance triggers the inherent reasoning capability of such models.

**Fine-tuning Learning.** As shown in Table 1, our DDCoT has achieved exceptional accuracy on fine-tuning benchmarks. DDCoT significantly surpasses the base model UnifiedQA [20] by 17.22% and 21.96% on avg split and IMG split, respectively. These demonstrate that our rationales not only enhance the model's multimodal reasoning ability in a zero-shot setting but also help the fine-tuning of

LMs to achieve multimodal alignment and joint inference. In comparison to the CoT-based approach MM-CoT [71] which uses annotated rationales for training, our multimodal rationales, generated through zero-shot CoT prompting, still enhance performance by an average of 2.43%.

| (a) Generalization | NAT | SOC | LAN |
|---|---|---|---|
| MMCoT | 50.98 | 77.73 | 80.00 |
| Ours | **66.43** | 83.69 | 83.09 |
| MMCoT | 78.86 | 62.20 | 76.82 |
| Ours | 86.5 | **71.77** | 85.09 |
| MMCoT | 79.53 | 76.38 | 55.82 |
| Ours | 86.59 | 86.73 | **68.00** |

| (b) Analysis | IMG | TXT | Avg |
|---|---|---|---|
| baseline(B) | 72.93 | 85.84 | 79.7 |
| B+caption | 75.26 | 83.64 | 79.6 |
| B+origin img | 62.82 | 75.94 | 69.70 |
| B+DLP&RCVE | **75.16** | **85.25** | **80.45** |
| no **R** | 75.16 | 85.25 | 80.45 |
| w/ naive **R** | 75.06 | 89.61 | 82.96 |
| w/ our **R** | **83.34** | **91.23** | **87.34** |

Table 2: The results of **(a) Generalization** and **(b)** Modality and Rationale **Analysis**.

**Generalization.** As shown in Table 2(a), we conduct a supplementary experiment to evaluate the model's generalization capabilities after fine-tuning. By designating two domains for visibility during training, we report the accuracy of the questions in the unseen remaining domain within the test set. Across all three divisions, our approach consistently surpasses MM-CoT by 15.5%, 9.6%, and 12.2%, respectively. These results demonstrate the excellent scalability of our rationales when handling out-of-distribution data.

## 4.2 Analysis Effects of Visual Modality, Rationale Generation, and Fine-tuning Components

**The effects of different modalities of visual information.** Table 2 (b) presents the effects of different forms of image information on the fine-tuning models: (1) We start by directly adding captions or image inputs to the language only T5 based baseline. The former results in an improvement of 2.33% on IMG split. However, the absence of additional components and the direct usage of images as input leads to poor performance. The result suggests that models with a confined number of parameters encounter difficulties aligning images with text on small datasets. (2) Nevertheless, with the aid of our well-designed components, including deep-layer prompting (DLP) and rational-compressed visual embedding (RCVE), the language model effectively harnesses the information gleaned from images, thereby achieving the comparable performance of 75.16% on IMG split with the model using captions as inputs. (3) Despite both methods enhancing performance on IMG split, the improvement remains unsatisfactory due to the language model's difficulty in comprehending image information without guidance, which is also observed in the zero-shot scenario delineated in Section 3.1. Simultaneously, we observe a slight degradation in the performance of the text split, which is potentially attributed to the disparity in inputs between IMG and TXT splits in training data.

**The effects of rationale generation components.** As shown in Table 2(b) and Table 3, we further demonstrate the effectiveness of the components of our DDCoT prompting for rationale generation. (1) With naive rationales obtained directly from prompting GPT-3 with *"let's think step by step"*, we noted that the IMG branch garners no benefits, even evidencing a slight degradation in performance.

| | IMG | Avg |
|---|---|---|
| naive **R** | 75.06 | 82.96 |
| decomposed question **R** | 75.61 | 83.09 |
| duty distinct w/o uncertainty **R** | 78.19 | 85.15 |
| duty distinct w/ uncertainty **R** | 83.34 | 87.34 |
| w/o integrate to joint reasoning | 77.49 | 83.75 |

Table 3: Ablation study on components of DDCoT.

It indicates that introducing rationales can facilitate the language model in comprehending and reasoning within the context, while image-agnostic rationales do not contribute to the model's multimodal understanding. (2) Compared to the naive rationale [25], deconstruction [72, 22] and joint reasoning without specialized designs result in a tiny improvement (row 2 in Table 3). The limited improvement can be attributed to the fact that LLMs still perform a combined reasoning and recognition task, leading to a suboptimal execution of the recognition task. (3) Hence, our duty distinct design, implemented through negative-space prompting, achieves 2.58% and 2.06% performance gain on IMG and average splits (row 3 in Table 3). It alleviates the reliance on the reasoning capability of the off-the-shelf visual model and generate more reliable multimodal rationales. (4) Moreover, by explicitly emphasizing uncertainty (row 4 in Table 3), we observe a notable improvement of 5.15% on IMG split, while also contributing to an average enhancement of 2.19%. It achieves remarkable enhancement of correctness, which also significantly contributes to fine-tuning learning.

**Quantitative Analysis of Hallucinations.**
Additionally, we conduct a human evaluation
of the authenticity to assess the phenomenon
of hallucinations [24] in the rationale gener-
ation process. As shown in the Table 4, in-
troducing interleaved visual information exac-
erbates the hallucinations and diminishes the

| Method | Context | Authenticity |
|---|---|---|
| naive **R** | w/o visual | 0.883 |
| naive **R** | interleaved | 0.602 |
| duty distinct w/o uncertainty **R** | interleaved | 0.783 |
| duty distinct w uncertainty **R** | interleaved | 0.855 |

Table 4: Human evaluation on hallucinations.

authenticity of generated rationales by 28.1% compared with naive prompting devoid of any visual
information input. Our duty-distinct design significantly mitigates the impact of hallucinations. More-
over, the further suppression of hallucinations is observed with the explicit emphasis on uncertainty.
These feedbacks align with the findings in Section 3.1.3 and the results presented in Table 3.

**The roles of our fine-tuning components.** We also vali-
date the impact of our proposed deep-layer prompting and
rational-compressed visual embedding. We replace the
rational-compressed embedding with origin visual features,
resulting in reductions of 1.28% and 3.02% in average per-
formance and IMG split, respectively. Similarly, removing
deep-layer prompting which assists alignment leads to a

| | IMG | Avg |
|---|---|---|
| Ours | **83.34** | **87.34** |
| w/o RCVE | 80.32 | 86.06 |
| w/o DLP&RCVE | 79.33 | 85.57 |

Table 5: Ablation study on fine-tuning
components.

0.49% and 0.99% reduction. Additionally, we further conduct ablation studies on the hyperparameters
of $N_p$ and $N_r$. Detailed results and analysis will be presented in the supplementary material.

| Method | w/o GT-**R** | Comparison with GT-**R** | | | | Human Evaluation | | | | |
|---|---|---|---|---|---|---|---|---|---|---|
| | | B-1 | B-4 | R-L | Sim. | Relevant | Correct | Complete | Coherent | Explainable |
| MMCOT | | 0.970 | 0.930 | 0.970 | 0.990 | 70.83% | 67.99% | 64.81% | 57.94% | 58.73% |
| GPT-3 | ✓ | 0.075 | 0.018 | 0.249 | 0.548 | 81.01% | 75.99% | 65.54% | 61.64% | 60.32% |
| Ours | ✓ | 0.147 | 0.041 | 0.287 | 0.601 | **92.00**% | **86.38**% | **85.71**% | **84.33**% | **83.26**% |

Table 6: Automatic metrics and human evaluation of explainability.

**Explainability.** We further exhibit the explainability evaluation including the automatic metrics
and human evaluations in Table 6. In automatic metrics, we adopt the BLEU-1/4 [38], ROUGE-
L [38], and Sentence Similarity [42] metrics. Note that the automatic evaluation solely reflects
the similarity between the generated rationales and the annotated ground truth. In the course of
human evaluation, annotators are required to grade each rationale on the criteria of Relevance,
Correctness, Completeness, Coherence, and Explainability. It is worth noting that although previous
methods [31, 71] achieve decent results in terms of Relevant and Correct, they perform particularly
poorly in other areas, especially Explainability. In contrast, despite modest automatic metric results,
our rationales generated by DDCoT significantly surpass other methods across all aspects of human
evaluation, which is more valuable for interpretable studies.

## 5 Conclusion

This paper proposes a novel DDCoT prompting for multimodal reasoning tasks with language models.
Not only achieving state-of-the-art performance on zero-shot learning, our model, with the novel
techniques of deep-layer prompting and rational-compressed visual embedding, also demonstrates
significant reasoning ability on the ScienceQA benchmark. The experimental results demonstrate
the superiority and generalization ability of our proposed DDCoT for both zero-shot learning and
fine-tuning. **Ethical Statement.** Our model is based on currently available pre-trained language
models. In our experiments, we have not utilized any information concerning personal privacy,
data security, or ethical hazards. However, it is crucial to acknowledge that large language models
may introduce certain biases, which can originate from the training datasets. We must recognize
that entirely eliminating biases from the model is a challenging task, and current technology may
not achieve it completely. Therefore, we encourage users to be mindful of these potential biases
when utilizing our model and to comprehend that the model's outputs are not infallible but require
interpretation and understanding in conjunction with real-world circumstances.

**Acknowledgment:** This work was supported by the National Natural Science Foundation of China
(No.62206174), Shanghai Pujiang Program (No.21PJ1410900), Shanghai Frontiers Science Center of
Human-centered Artificial Intelligence (ShangHAI), MoE Key Laboratory of Intelligent Perception
and Human-Machine Collaboration (ShanghaiTech University), and Shanghai Engineering Research
Center of Intelligent Vision and Imaging.

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

# Appendix

In this section, we present additional implementation details, experiment results and discussions. The content structure is outlined as follows:

# A  Additional Results

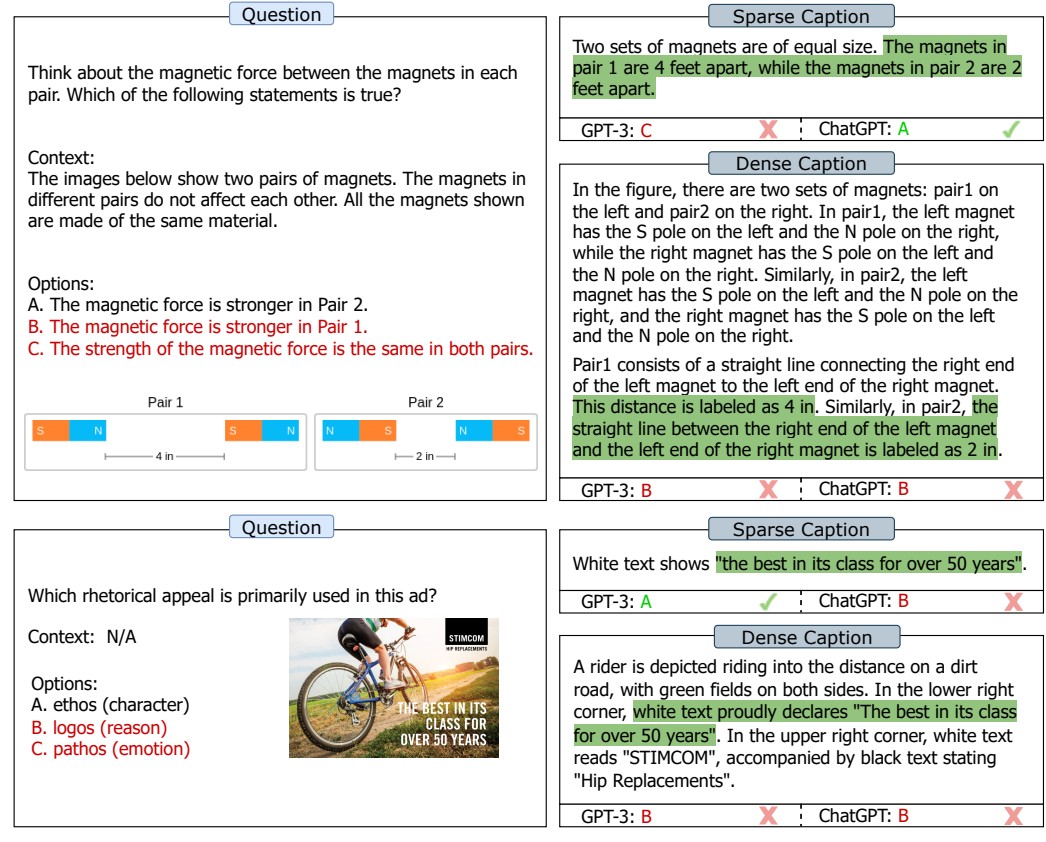

Figure 6: Cases that GPT-3 and ChatGPT face the difficulty in understanding dense image information.

## A.1  Insights for Different GPT Models

In the submitted paper we have presented several findings concerning the LLMs. In this section, we aim to provide additional illustrative instances for GPT-3 [3] and the recent and potent ChatGPT [37].

**Difficulty in understanding dense image information.** Figure 6 presents additional instances of LLMs failing to comprehend dense image information. For sparse captions like the example in Figure 2 of the submitted paper, we observe that both GPT-3 and ChatGPT may struggle to comprehend image information in captions. Additionally, when prompted with more detailed captions, their difficulties in understanding become more pronounced. This highlights the challenges that even the most versatile and powerful language models currently available face in comprehending dense image information.

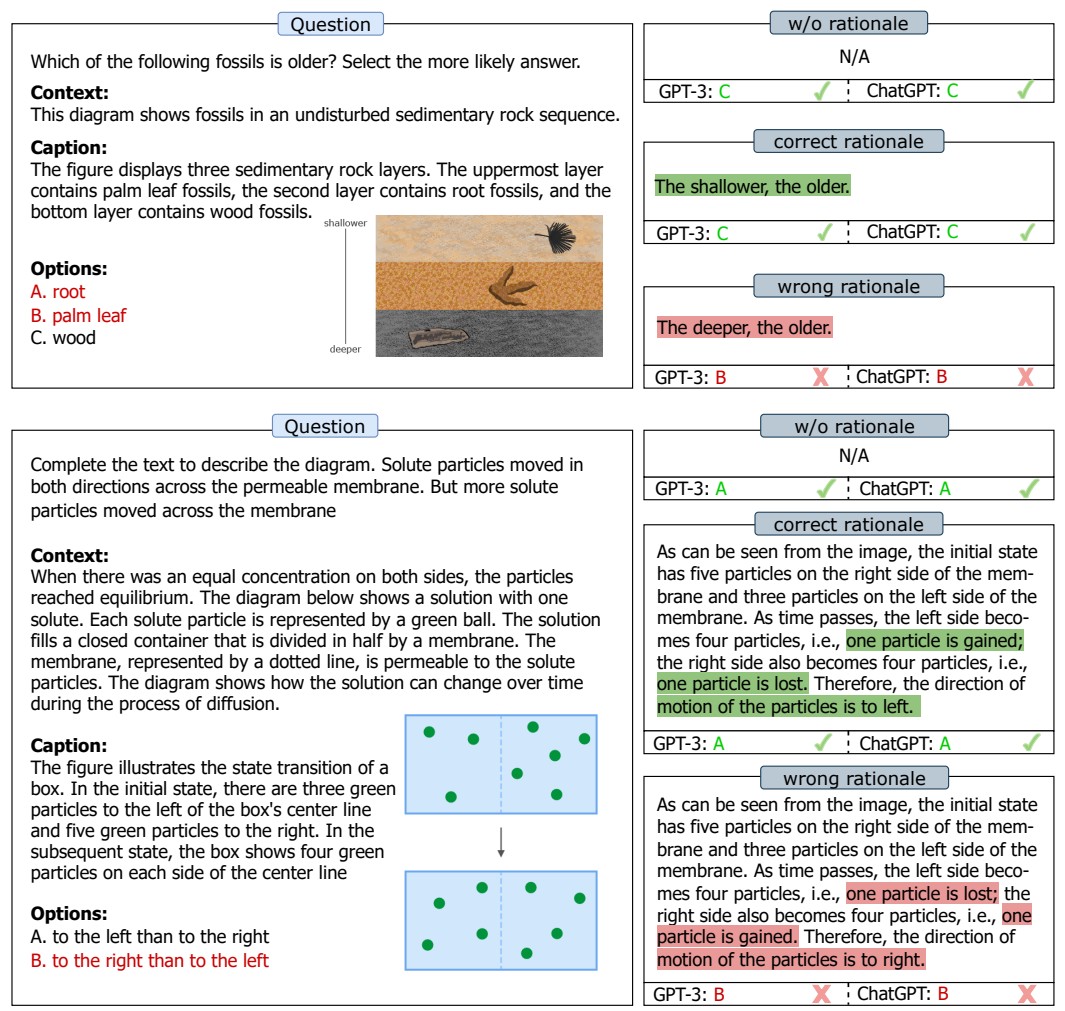

Figure 7: Cases of rationale-sensitive reasoning with GPT-3 and ChatGPT.

**Rationale-sensitive reasoning.** Figure 7 shows more examples where the reasoning of GPT-3 and ChatGPT is sensitive to the input rationales. The correct answers obtained without using rationales indicate that the LLMs possess commonsense knowledge to respond to the question. However, incorrect inputs can lead to misleading outcomes for both GPT-3 [3] and ChatGPT [37].

Note that the challenges involved in understanding dense image information, coupled with the difficulty of obtaining high-quality captions in practical scenarios, limit the universal applicability of reasoning without using rationales, although it may be feasible in the presented examples.

**Hallucinations during generation.** Figure 8 illustrates several instances where both GPT-3 and ChatGPT fall into hallucinations during rationale generation.

Our findings indicate that in cases where the provided caption information is insufficient, such as in the first example, both GPT-3 [3] and ChatGPT [37] tend to imagine image details to respond to the question. Even when we provide ample information, as demonstrated in the second and third

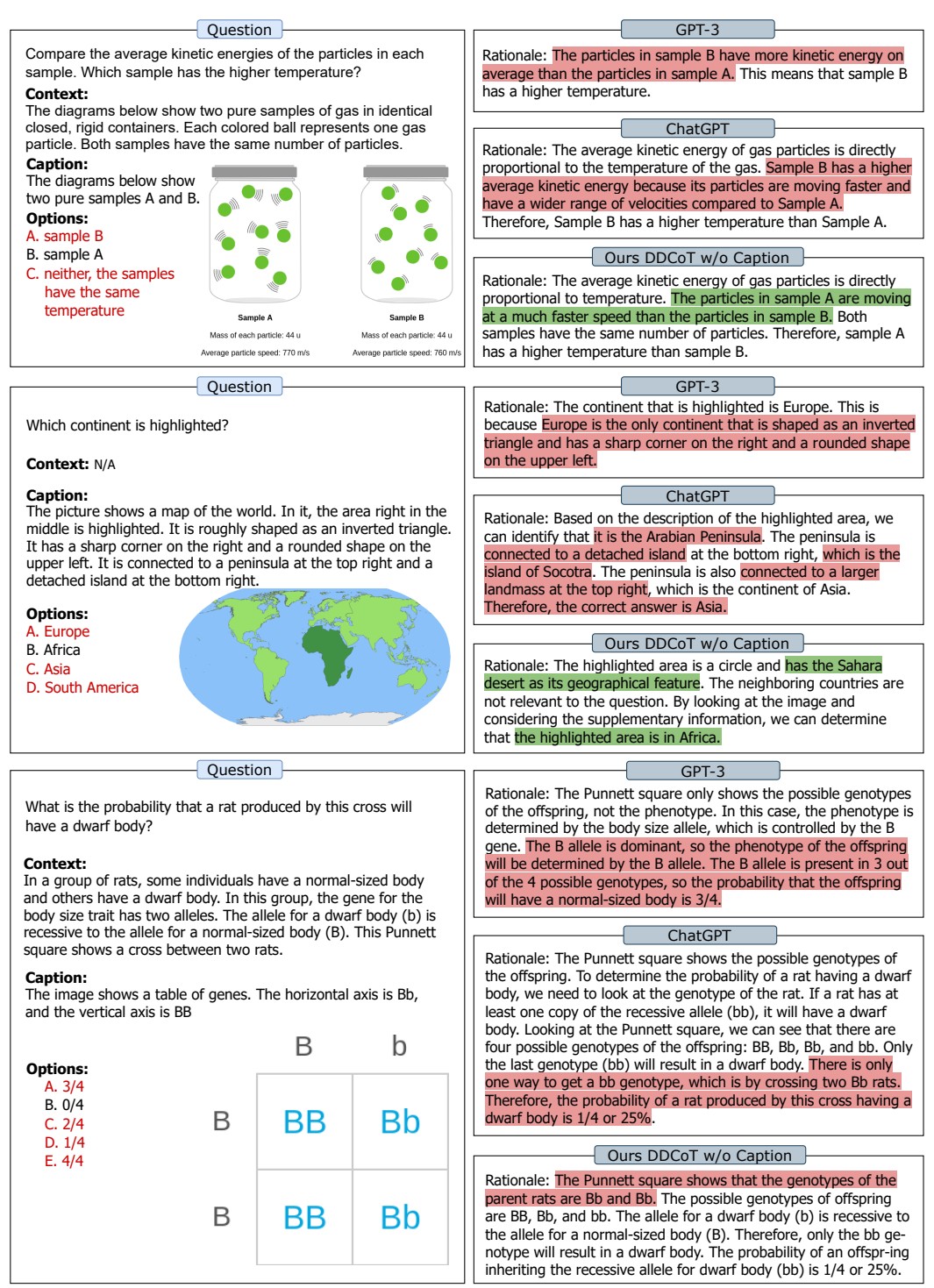

Figure 8: Cases of hallucinations when generating rationales with GPT-3 and ChatGPT.

examples, we observe that both language models still may fall into hallucinations that do not align with the provided information.

In contrast to generating rationales solely based on the caption and question information, our approach can alleviate the hallucinations to some extent by decomposing the questions into simple recognition tasks and emphasizing the uncertainty of image-related aspects. It is worth noting that while the former direct methods rely on manually crafted high-quality rationales, our approach utilizes BLIP-2 [26] as a visual question answering (VQA) model, serving as a visual component. However, our approach outperforms the former methods in terms of performance and interpretability.

Unfortunately, it is very challenging to entirely solve the issues of hallucinations. Although we alleviate hallucinations, we still encountered difficulties in certain cases, such as the third example, and further details regarding this limitation will be explored in section E.

|  | IMG | TXT | Avg |
|---|---|---|---|
| baseline(B) | 72.93 | 85.84 | 79.70 |
| B + our R | 75.81 | 82.40 | 82.83 |
| B + gt R | 81.07 | 84.07 | 85.97 |
| our model | 75.16 | 85.25 | 80.45 |
| our model + our R | 83.34 | 91.20 | 87.34 |
| our model + gt R | 84.43 | 92.09 | 88.00 |

Table 7: Quantitative ablations on our DDCoT and visual components.

## A.2 Quantitative ablations on our DDCoT and visual components

We conduct additional ablation study on the extent of impact exerted by our DDCoT prompting and visual components, as shown in Table 7. DDCoT prompting and visual components cooperatively facilitate inducing visual information to language models for multimodal reasoning. We can observe that rationales generated by our DDCoT and visual components individually exhibit certain gains in terms of IMG improvement. However, when combined, they yield substantial gains.

Besides, please note that the annotated ground truth rationales within the ScienceQA dataset inherently encompass the final prediction, i.e., correct answers. To ensure a fair comparison, we manually exclude the answers from these annotations, using the remaining text as input rationales for fine-tuning. Under this configuration, our proposed rationale achieves a fine-tuning performance comparable to the annotated rationales.

| $N_p$ | 1 | 3 | 5 | $N_r$ | 8 | 16 | 32 |
|---|---|---|---|---|---|---|---|
| Avg | 86.72 | **87.34** | 86.02 | Avg | 86.63 | **87.34** | 86.30 |

Table 8: Ablation of $N_p$ and $N_r$.

## A.3 Hyperparameters for Fine-tuning Components

Table 8 presents the results of our ablation studies on $N_p$, which denotes the number of learnable prompt tokens, and $N_r$, which represents the number of low-rank vectors. Regarding $N_p$, we conducted experiments using values of 1, 3, and 5. Increasing the number of prompts introduces more learnable parameters and provide stronger guidance to align vision and language, while too many prompts may disrupt the model's comprehension of visual features and result in a decline in performance. Considering $N_r$, we investigated the influence of different filtering intensity in the Rational-Compressed Visual Embedding process by experimenting with values of 8, 16, and 32. The experimental results indicate that selecting 3 for $N_p$ and 16 for $N_r$ yields the best performance.

## A.4 Additional experiments on the effectiveness of our DDCoT with existing pre-trained VLMs and multimodal reasoning models

We also validate the effectiveness of our DDCoT with existing pre-trained VLMs and multimodal reasoning models, as shown in Table 9. We observe that rationales generated by our proposed DDCoT is compatible with such pre-trained VLMs. Without correct rationales, existing pretrained

| Model | NAT | SOC | LAN | TXT | IMG | NO | Avg |
|---|---|---|---|---|---|---|---|
| Flamingo [2] | 21.89 | 52.41 | 20.27 | 23.50 | 39.11 | 19.02 | 27.87 |
| Flamingo with our R | 39.20 | 48.93 | 30.90 | 39.68 | 45.81 | 32.40 | 39.01 |
| MiniGPT-4 [75] | 43.83 | 48.59 | 43.36 | 55.01 | 42.84 | 41.67 | 44.71 |
| MiniGPT-4 with our R | 57.37 | 62.32 | 46.82 | 65.91 | 56.72 | 48.57 | 55.67 |

Table 9: The effectiveness of our DDCoT with Flamingo [2] and MiniGPT-4 [75]

VLMs and multimodal reasoning models have difficulty in complex reasoning tasks. Fortunately, the generalizable rationales generated by our DDCoT prompting can help existing VLMs to comprehend visual information and reason with rich knowledge, achieving significant improvement of 11.14% and 10.96% based on Flamingo [2] and Mini GPT-4 [75] as Table 9 shows.

| | NoCaps | | | | MSVD-QA |
|---|---|---|---|---|---|
| | CIDEr | SkipThoughtCS | EmbeddingAverageCS | GreedyMatchingScore | Acc |
| BLIP-2 | 76.15 | 49.84 | 89.20 | 77.94 | 34.4 |
| Ours | 46.26 | 84.78 | 92.35 | 79.12 | 39.3 |

Table 10: Addition experiments on NoCaps and MSVD-QA.

## A.5 Additional experiments on Captioning and Video Question Answering tasks

We extend our approach to more appropriate datasets. While other existing datasets may not fully exploit the benefits of our approach, we venture into exploring the captioning task on NoCaps [1] and the video question-answering task on MSVD-QA [64], as shown in Table 10.

For captioning, we prompt the LLM [37] to solve sub-problems derived from a simple caption, aiming to optimize and enrich it using the corresponding sub-answers. The substantial knowledge within LLM enables the generation of semantically enriched captions, leading to improvements in metrics evaluating sentence semantics, i.e. 34.94%, 3.15%, and 1.18% in terms of SkipThoughtCS [23], EmbeddingAverageCS [46], and GreedyMatchingScore [45]. Note that the CIDEr [57] metric to evaluate ours is limiting. It is designed to measure the similarity between the tested caption and reference captions without considering the diversity and high-level semantics.

For video question answering, LLM deconstructs problems like the decomposition step on ScienceQA. We sample video frames for VQA recognition and integrate frame information for multimodal rationale and answers. Leveraging the sequence understanding in LLM and visual information returned by the VQA model, we achieve a 4.9% improvement over BLIP-2 [26].

Note that we randomly evaluated only 1000 images from NoCaps and 1000 videos from the MSVD test dataset in a zero-shot setting.

# B Human Evaluation

This section introduce the details of our human evaluation. Figure 9(b) shows an example of our question. Each sample comprises the question, context, options, and image. Evaluators are asked to rate the rationales generated by GPT [3], MM-COT [71], and our method in five aspects: relevance (related to the question), correctness (accuracy of reasoning and answer), completeness (logical reasoning's comprehensiveness), coherence (consistency of reason), and explainability (interpretability of reasoning and answer). The rating scale ranges from 0 to 5. Additionally, the questions and rationales are organized into 12 groups, with each group assigned to three evaluators. Finally, we average the scores for each aspect of each rationale, resulting in overall scores and ratios relative to the maximum score.

# C Detailed Prompts

This section introduce more details about our zero-shot DDCoT prompting. We employ ChatGPT[37] as the most important component of our rationale generator. Specifically, it is utilized in breaking

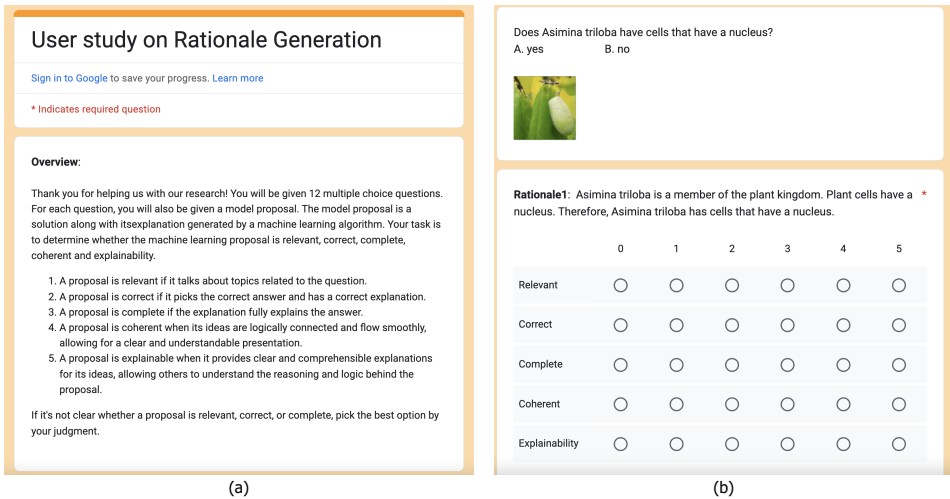

Figure 9: Interface of human evaluation. Figure(a) shows the instructions, figure(b) shows one example of our question.

duties of reasoning and recognition step and joint reasoning step. Figure 10 shows complete prompts for zero-shot DDCoT prompting.

# D    Case Studies

To better understand the effectiveness of our proposed method in generating rationales, we randomly selected several cases from the test set along with the process of rationale generation. Figure 11 showcases several map-related questions, demonstrating how our method integrates simple visual features (such as the shape of highlighted areas) with common knowledge to obtain correct reasoning and answers. In the examples presented in Figure 12, our method successfully identifies within the images, acquiring relevant knowledge. Figure 13 illustrates four more complex questions, where our method leverages information obtained from the images to perform intricate reasoning. However, when it comes to the complex interaction between images and textual context, our method still fall into hallucinations, leading to erroneous reasoning and answers.

# E    Limitations

While we have succeeded in mitigating a portion of the hallucination problem arising from multimodal inputs, it is important to acknowledge that this problem is not entirely resolved. As illustrated in Figure 13, our model is also susceptible to the risk of hallucinations. Investigating methods to suppress hallucinations is a potential topic for further research and exploration. In addition, we did not use extra image-text pairs to pre-train the alignment between vision and language modalities. Such pre-trianing is expected to further improve the alignment for joint reasoning.

As our approach involves zero-shot prompting of the LLM to generate rationales, there exists a potential risk of inheriting social biases from the LLM. These biases, which encompass cultural, ethical, and various other dimensions, might be reflected in the generated rationales, potentially leading to adverse effects on users. To mitigate this issue in the future, potential solutions could involve designing constraints at each prompting stage or utilizing more advanced LLMs trained on unbiased resources.

| DDCoT |
|---|

**Breaking Reasoning Down to Recognition Steps with Negative-Space Prompting**

**System:** You are a helpful, highly intelligent guided assistant. You will do your best to guide humans in choosing the right answer to the question. Note that insufficient information to answer questions is common, because you do not have any information about the picture. The final answer should be one of the options.

**User:** Given the context, questions and options, please think step-by-step about the preliminary knowledge to answer the question, deconstruct the question as completely as possible down to necessary sub-questions based on context, questions and options. Then with the aim of helping humans answer the original question, try to answer the sub-questions. The expected answering form is as follows:
Sub-questions:
1. <sub-question 1>
2. <sub-question 2>
...
Sub-answers:
1. <sub-answer 1> or 'Uncertain'
2. <sub-answer 2> or 'Uncertain'
...
Answer: <One of the options> or 'Uncertain'

For a question, assume that you do not have any information about the picture, but try to answer the sub-questions and prioritize whether your general knowledge can answer it, and then consider whether the context can help. If sub-questions can be answered, then answer in as short a sentence as possible. If sub-questions cannot be determined without information in images, please formulate corresponding sub-answer into "Uncertain".
Only use "Uncertain" as an answer if it appears in the sub-answers. All answers are expected as concise as possible.
Here is an attempt:
Context: {context}
Has An Image: {has_image}
Question: {question}
Options: {option}"

- - - - - - - - - - - - - - - - - - - - - - - - - - - - - - - - - - - - - - - - - - - - - - - - - - - - - - - - - -

**Visual Recognition to Obtain Visual Complements**

~ Aribitrary Visual Question Answering (VQA) Model ~

- - - - - - - - - - - - - - - - - - - - - - - - - - - - - - - - - - - - - - - - - - - - - - - - - - - - - - - - - -

**Integrate to Joint Reasoning**

**System:** You are a helpful, highly intelligent teacher. You will not only do your best to guide humans to the correct answer, but you will also give the rationales as a reference.

**User:** Given the context, questions, options, supplementary information, think step by step and answer the questions. Please note that we need not only the answer, but more importantly the rationales of getting the answer. The expected answering form is as follows:
Rationale: <rationale>
Answers: <one of the options>

Please note that the supplementary information given may not always be valid. Please select valid information to form the rationale and choose the relatively correct option as your answer.
Here is an attempt:
Context: {context}
Has An Image: {has_image}
Question: {question}
Options: {option}
Supplementary information: {supplementary_information}

Figure 10: The complete prompts of our zero-shot DDCoT prompting. The underlined parts are introduced in Section 3.2 of the submitted paper. The yellow background represents designs for critical thinking, and the blue background represents designs for differentiate duties of reasoning and recognition. "{}" denotes the corresponding inputs, and "{supplymentary_information}" represents the sub-questions and sub-answers obtained before.

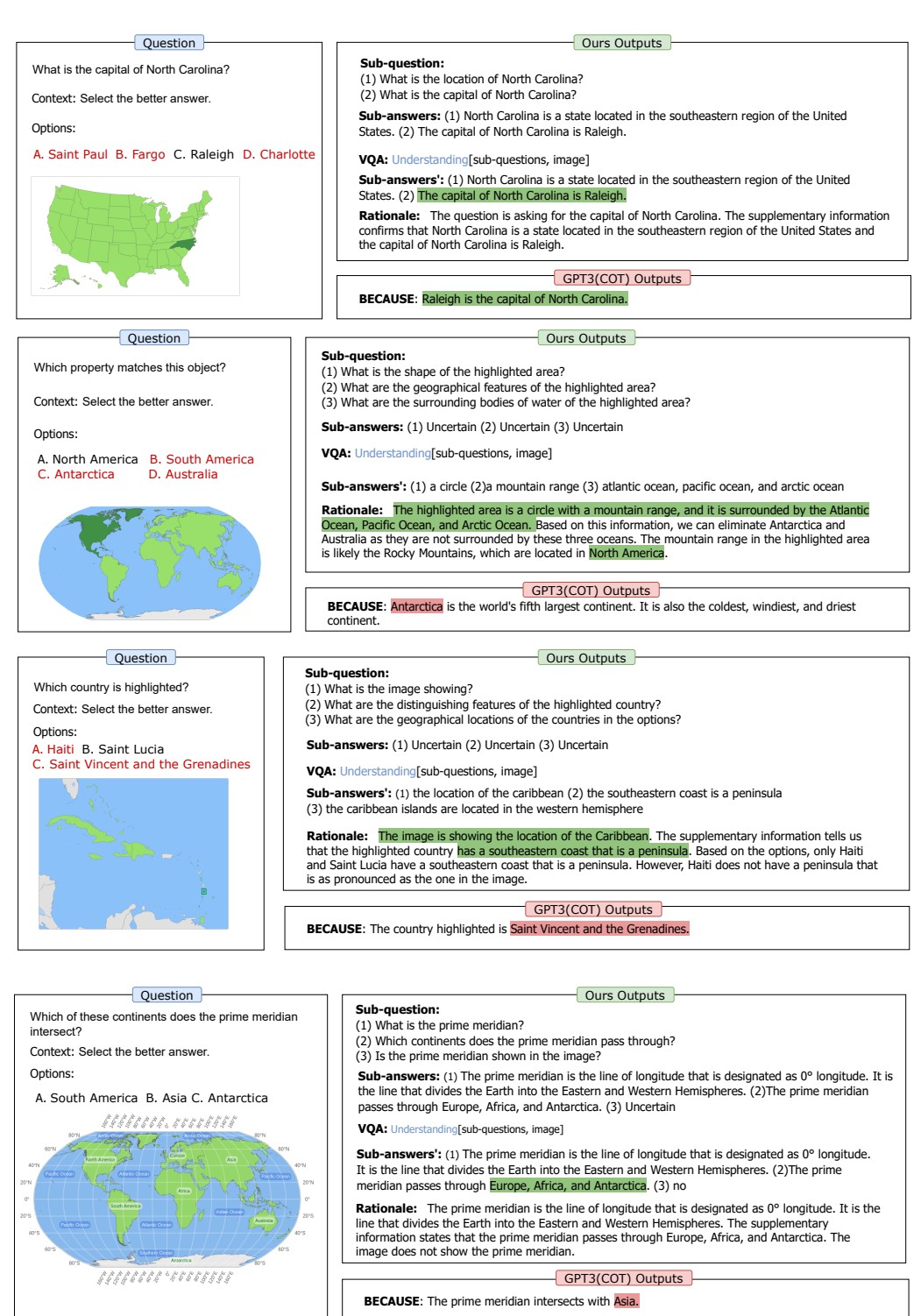

Figure 11: Cases that need to understand the map.

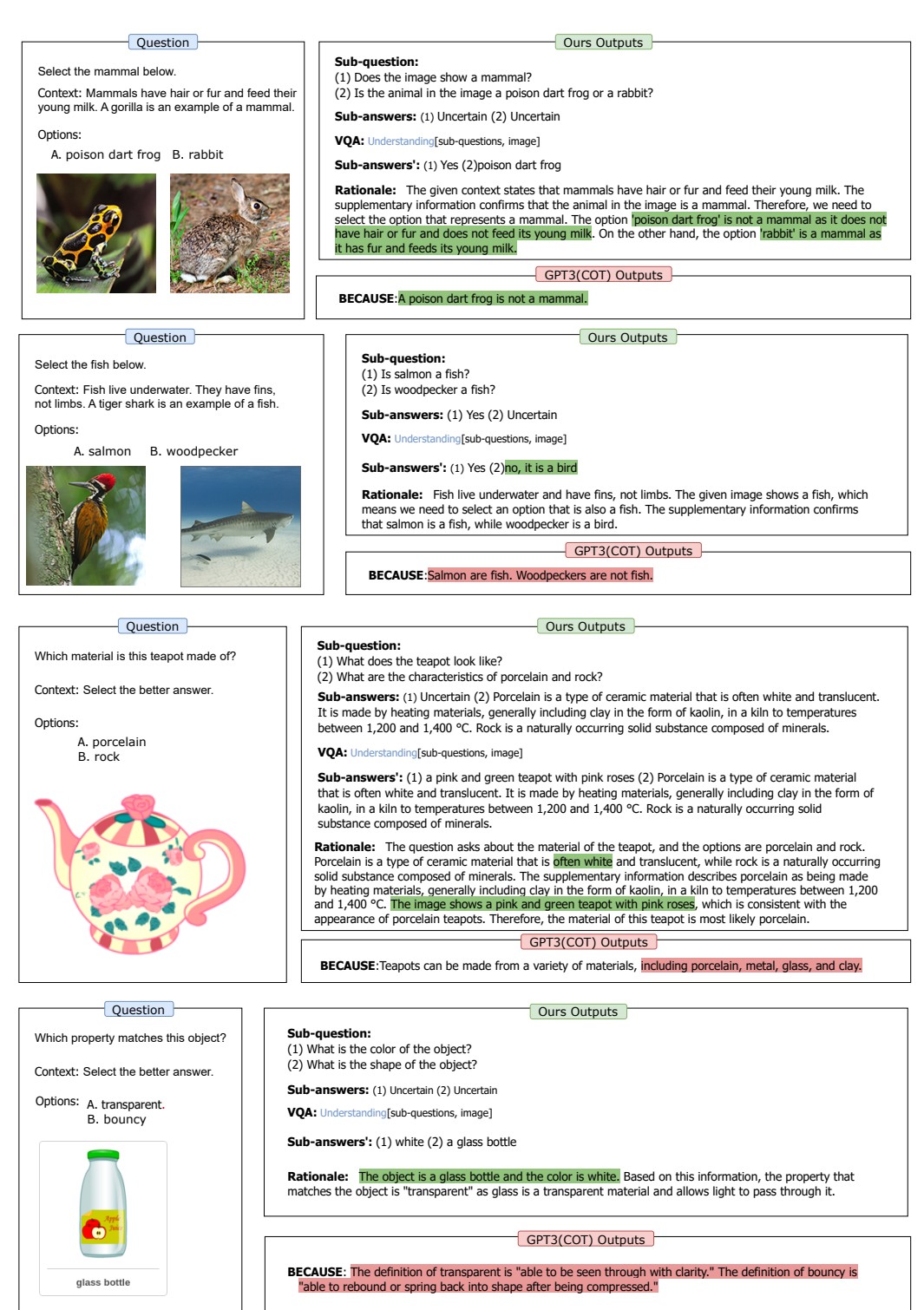

**Question**

Select the mammal below.

Context: Mammals have hair or fur and feed their young milk. A gorilla is an example of a mammal.

Options:

A. poison dart frog    B. rabbit

**Ours Outputs**

**Sub-question:**
(1) Does the image show a mammal?
(2) Is the animal in the image a poison dart frog or a rabbit?

**Sub-answers:** (1) Uncertain (2) Uncertain

**VQA:** Understanding[sub-questions, image]

**Sub-answers':** (1) Yes (2)poison dart frog

**Rationale:** The given context states that mammals have hair or fur and feed their young milk. The supplementary information confirms that the animal in the image is a mammal. Therefore, we need to select the option that represents a mammal. The option 'poison dart frog' is not a mammal as it does not have hair or fur and does not feed its young milk. On the other hand, the option 'rabbit' is a mammal as it has fur and feeds its young milk.

**GPT3(COT) Outputs**

**BECAUSE:**A poison dart frog is not a mammal.

**Question**

Select the fish below.

Context: Fish live underwater. They have fins, not limbs. A tiger shark is an example of a fish.

Options:

A. salmon    B. woodpecker

**Ours Outputs**

**Sub-question:**
(1) Is salmon a fish?
(2) Is woodpecker a fish?

**Sub-answers:** (1) Yes (2) Uncertain

**VQA:** Understanding[sub-questions, image]

**Sub-answers':** (1) Yes (2)no, it is a bird

**Rationale:** Fish live underwater and have fins, not limbs. The given image shows a fish, which means we need to select an option that is also a fish. The supplementary information confirms that salmon is a fish, while woodpecker is a bird.

**GPT3(COT) Outputs**

**BECAUSE:**Salmon are fish. Woodpeckers are not fish.

**Question**

Which material is this teapot made of?

Context: Select the better answer.

Options:
A. porcelain
B. rock

**Ours Outputs**

**Sub-question:**
(1) What does the teapot look like?
(2) What are the characteristics of porcelain and rock?

**Sub-answers:** (1) Uncertain (2) Porcelain is a type of ceramic material that is often white and translucent. It is made by heating materials, generally including clay in the form of kaolin, in a kiln to temperatures between 1,200 and 1,400 °C. Rock is a naturally occurring solid substance composed of minerals.

**VQA:** Understanding[sub-questions, image]

**Sub-answers':** (1) a pink and green teapot with pink roses (2) Porcelain is a type of ceramic material that is often white and translucent. It is made by heating materials, generally including clay in the form of kaolin, in a kiln to temperatures between 1,200 and 1,400 °C. Rock is a naturally occurring solid substance composed of minerals.

**Rationale:** The question asks about the material of the teapot, and the options are porcelain and rock. Porcelain is a type of ceramic material that is often white and translucent, while rock is a naturally occurring solid substance composed of minerals. The supplementary information describes porcelain as being made by heating materials, generally including clay in the form of kaolin, in a kiln to temperatures between 1,200 and 1,400 °C. The image shows a pink and green teapot with pink roses, which is consistent with the appearance of porcelain teapots. Therefore, the material of this teapot is most likely porcelain.

**GPT3(COT) Outputs**

**BECAUSE:**Teapots can be made from a variety of materials, including porcelain, metal, glass, and clay.

**Question**

Which property matches this object?

Context: Select the better answer.

Options: A. transparent.
B. bouncy

glass bottle

**Ours Outputs**

**Sub-question:**
(1) What is the color of the object?
(2) What is the shape of the object?

**Sub-answers:** (1) Uncertain (2) Uncertain

**VQA:** Understanding[sub-questions, image]

**Sub-answers':** (1) white (2) a glass bottle

**Rationale:** The object is a glass bottle and the color is white. Based on this information, the property that matches the object is "transparent" as glass is a transparent material and allows light to pass through it.

**GPT3(COT) Outputs**

**BECAUSE:** The definition of transparent is "able to be seen through with clarity." The definition of bouncy is "able to rebound or spring back into shape after being compressed."

Figure 12: Cases that need to identify objects and acquire relevant knowledge.

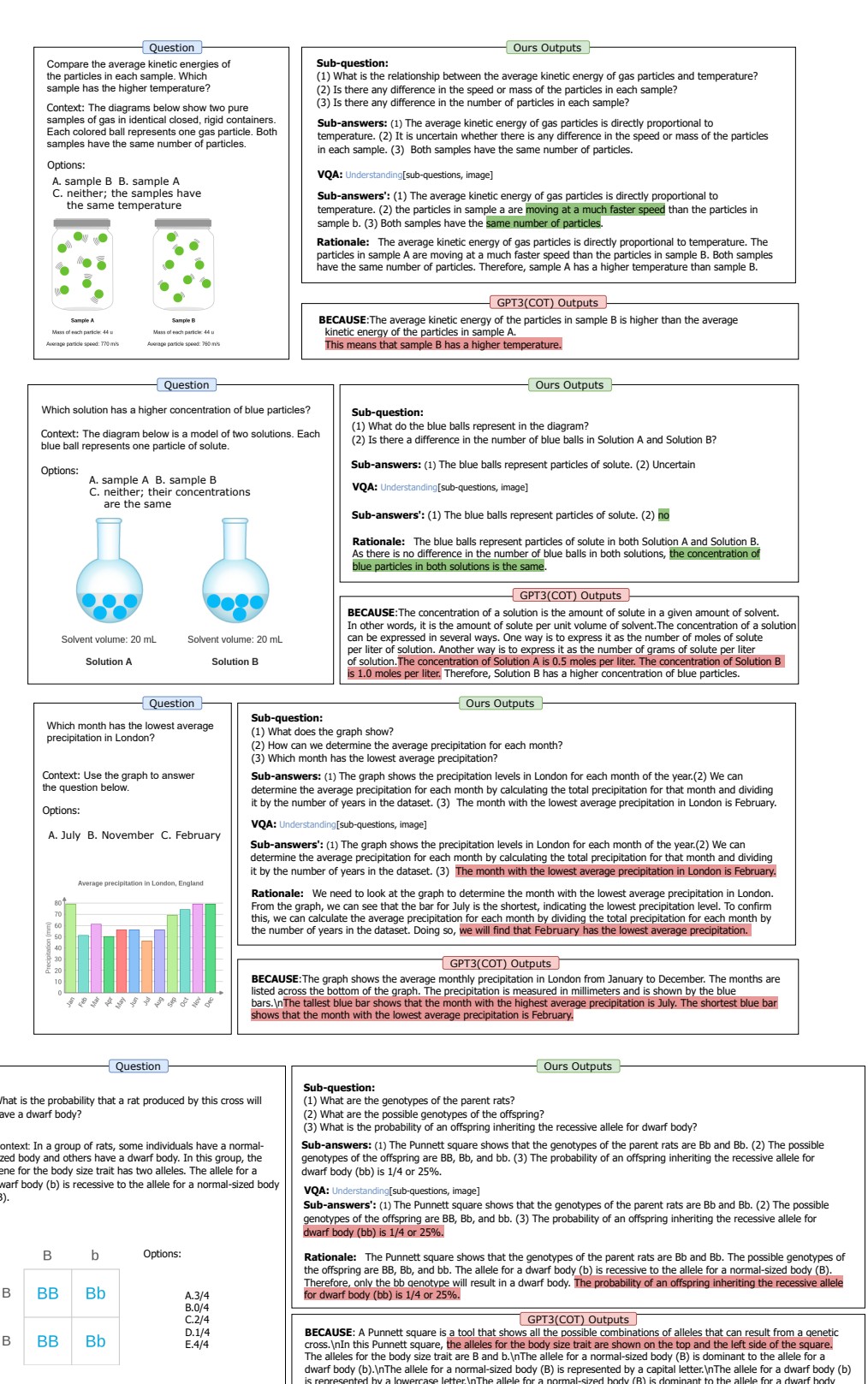

Figure 13: Complex and Challenging cases.

