# OpenReview forum: "DDCoT: Duty-Distinct Chain-of-Thought Prompting for Multimodal Reasoning in Language Models"
_NeurIPS.cc/2023/Conference — NeurIPS 2023 poster_

### Official Review · Reviewer_ndag · 2023-06-12

**Soundness:** 2 fair
**Presentation:** 3 good
**Contribution:** 2 fair
**Rating:** 5
**Confidence:** 4

**Summary:**

This paper explore utilizing the rationales to achieve the multimodal reasoning based on LMs.
The author analyzes the challenges on using the LM to perform the multimodal reasoning such as Hallucination problem.
Based on the preliminary observations, the authors propose jointly exploit the reasoning ability in LLMs and the image understanding capability of visual question-answering models for general multimodal rationale generation.
Experimental results demonstrate that the proposed method named DDCoT is effective from the experimental results.

**Strengths:**

This paper clearly expresses their motivation and analyzes the common problems when exploiting language models for multimodal reasoning.
1. The motivation is clear and the analysis of hallucination problem is comprehensive.
2. The performance of the proposed DDCoT is demonstrated to be superior to some common baselines.

**Weaknesses:**

The method proposed in this paper seems to be a simple performance combination of large language model and existing visual question answering model, and mainly utilizes the ability of large language model to decompose questions. Such a kind of method has already been explored in the research direction of using large language models to call external tools, such as Visual Chatgpt, Augmented Language Models.
1. Although the logic of the proposed method is clear, the technical novelties seem to be insufficient. The proposed method is similar to the Self-asking COT, or ``Demonstrate-Search-Predict: Composing retrieval and language models for knowledge-intensive NLP’’. For the multimodal information interaction, this paper does not present the detailed description.
2. The claims in lines 75-76 show that this paper is the first to explore the zero-shot multimodal reasoning. However, they do not give the enough analysis of this point and do not consider the performance of pretrained VLMs such as OFA or Flamingo. In addition, similar methods such as ModCR[1], Mini GPT-4, BLIP-2, and others should be compared.
3. The proposed module Rational-Compressed Visual Embedding (RCVE) aims to compress visual input embeddings according to the multimodal rationales by filtering visual features. I am confused by this calculation. For example, Formula 1, it adopts global visual features as query vectors and rational information as Key and Value. This process is the integration of different information, which could be not in line with its purpose.
4. Deep-Layer Prompting (DLP) is similar to the Prompt tuning v2 method [2]. The whole approach should be improved. Missing the corresponding citation.
[1] Liu X, Ji K, Fu Y, et al. P-tuning v2: Prompt tuning can be comparable to fine-tuning universally across scales and tasks[J]. arXiv preprint arXiv:2110.07602, 2021.
[2] Guo J, Li J, Li D, et al. From Images to Textual Prompts: Zero-shot VQA with Frozen Large Language Models[J]. arXiv preprint arXiv:2212.10846, 2022.

**Questions:**

See the above strength and weakness parts

---

> ### Author Rebuttal · Authors · 2023-08-10
>
> We thank the reviewer for the feedback. Please find our responses below.
>
> > **Response to the overall weaknesses:**
> >
>
> We thank the reviewer for the rigorous consideration. But we have to claim that our contributions and technical novelties are far beyond the “simple combination of LLMs and VQA models” (simple combination) and the “utilization of the ability of LLMs to decompose questions” (simple decomposition):
>
> 1. **We are the first to (1)** study and achieve zero-shot multimodal CoT rationale generation, **(2)** discover LLMs’ hallucinations are intensified with interleaved multimodal inputs and thus deeply analyze how to put visual information in the text to generate helpful rationales, and **(3)** propose a novel prompting method (DDCoT) to generate rationales works for both zero-shot and finetuning learning. Previous works and the mentioned Visual ChatGPT and Augmented Language Models do not explore these aspects.
> 2. **The above simple combination and decomposition cannot achieve multimodal rationale generation**.
>     1. Such a simple combination predisposes LLMs towards hallucination, generating fabricated rationales (see Lines 172-183). As a result, rationales simply generated from LLMs with captions offer significantly constrained assistance for multimodal reasoning (73.33 on IMG and 82.15 on average) in comparison to our approach (83.34 on IMG and 87.34 on average).
>     2. Also, the simple decomposition without considering “negative-space prompting” (Lines 194-205) or “integrating to joint reasoning” (Lines 214-223) cannot work well, i.e., -7.73% (Lines 15-19 in the Appendix) for the former, and -5.85% (refer to the respond table to the Reviewer ka68) for the latter on the IMG split, in comparison to our DDCoT.
> 3. Our DDCoT not only **significantly outperforms** previous methods but also exhibits impressive generalization ability and explainability.
>
> > **Technical novelties** seem to be insufficient. The proposed method is similar to the Self-asking COT, or Demonstrate-Search-Predict.
> >
>
> Regarding the concern about our novelties, please refer to our “response to the overall weaknesses”.
>
> The significant differences compared to Self-asking COT and Demonstrate-Search-Predict are:
>
> 1. **Focuses and motivations:** The Self-asking COT and Demonstrate-Search-Predict work solely on the language modality, and they have different focuses compared to our multimodal CoT rationale generation. In contrast, the new challenges and corresponding solutions brought by multimodality are what we focus on.
> 2. **Technicals**: Self-asking COT and Demonstrate-Search-Predict have not explored at least two of our core ideas: (1) distinct duties by considering the uncertainty, (2) integrating sub-question and answers to generate one coherent rationale instead of final prediction.
>
> > **Multimodal reasoning:** The claims in lines 75-76 show that this paper is the first to explore the zero-shot multimodal reasoning. However, they do not give the enough analysis of this point and do not consider the performance of pretrained VLMs such as OFA or Flamingo. In addition, similar methods such as ModCR[1], Mini GPT-4, BLIP-2, and others should be compared.
> >
> 1. This work is the first to study the “zero-shot **multimodal rationale generation”** (Lines 75-76) rather than “multimodal reasoning”.
> 2. In fact, our proposed rational generation is compatible with such pretrained VLMs and methods of multimodal reasoning. Without correct rationales, existing pretrained VLMs and multimodal reasoning models have difficulty in complex reasoning tasks. Fortunately, **the generalizable rationales generated by our DDCoT prompting can help existing VLMs** to comprehend visual information and reason with rich knowledge, achieving significant improvement of **11.14**% and **10.96**% based on Flamingo and Mini GPT-4 as the below table shows. Thanks for your feedback, and we will include the analysis and experiments.
>
>
>     |  | NAT | SOC | LAN | TXT | IMG | NO | Avg |
>     | --- | --- | --- | --- | --- | --- | --- | --- |
>     | OFA | 5.91 | 0.11 | 13.18 | 13.05 | 0.30 | 11.85 | 6.58 |
>     | Flamingo | 21.89 | 52.41 | 20.27 | 23.50 | 39.11 | 19.02 | 27.87 |
>     | Flamingo with our R | 39.20 | 48.93 | 30.90 | 39.68 | 45.81 | 32.40 | 39.01 |
>     | Mini GPT-4 | 43.83 | 48.59 | 43.36 | 55.01 | 42.84 | 41.67 | 44.71 |
>     | Mini GPT-4 with our R | 57.37 | 62.32 | 46.82 | 65.91 | 56.72 | 48.57 | 55.67 |
>     | BLIP-2 | 67.40 | 56.36 | 52.45 | 67.01 | 62.77 | 53.58 | 61.21 |
>     | Ours | 88.72 | 86.84 | 84.91 | 87.59 | 83.34 | 88.08 | 87.34 |
>
> > **Calculation in RCVE:** The proposed module Rational-Compressed Visual Embedding (RCVE) aims to compress visual input embeddings according to multimodal rationales by filtering visual features. I am confused by this calculation.
> >
>
> Formula 1 and Formula 2 work together to form the RCVE module. Global visual features are updated based on the similarity to the input rationales in Formula 1. In formula 2, the updated visual features first reshape to $N_r$  low-rank intermediate vectors and then capture local visual information through the attention mechanism. These processes culminate in the compression of local visual information into $N_r$ visual embeddings.
>
> > **Citation:** Deep-Layer Prompting (DLP) is similar to the Prompt tuning v2 method [2]. The whole approach should be improved. Missing the corresponding citation [1][2].
> >
>
> Thanks for your valuable feedback. We will cite and discuss the shallow prompting [1,2,3,4] and deep prompting [5,6,7,8] methods, which are common in prompting learning. Our DLP is used to facilitate the alignment of visual and linguistic semantics at a shallow level and to combine with RCVE to utilize rationale to encode multimodality jointly at each layer.
>
> See our response to Reviewer aDZq for the citations [1-8].

---

> > ### Comment · Reviewer_ndag · 2023-08-22
> > **Official Comments from ndag**
> >
> > The rebuttal addresses my comments. The paper does have merits. Therefore, I am now tending to accept the paper.

---

### Official Review · Reviewer_aDZq · 2023-06-29

**Soundness:** 3 good
**Presentation:** 4 excellent
**Contribution:** 3 good
**Rating:** 6
**Confidence:** 4

**Summary:**

This paper proposes a novel DDCoT prompting that maintains a critical attitude through negative-space prompting and incorporates multimodality into reasoning by first dividing the reasoning responsibility of LLMs into reasoning and recognition and then integrating the visual recognition capability of visual models into the joint reasoning process.

**Strengths:**

- The authors deeply analyze the challenges and insights in multimodal CoT for the rationale generation.
- The authors propose a novel DDCoT prompting to maintain a critical attitude and identify reasoning and recognition responsibilities  through the combined effect of negative-space design and deconstruction.
- The experiments show the superiority of the proposed method.

**Weaknesses:**

- Are the analysis results of the single case  in section 3.1.1 and 3.1.2  applicable to most samples in the data set? In addition to chatgpt, whether gpt-3 can reach similar conclusions?
- On what basis are the three principles in 3.2 considered？
- Why the “uncertainty”  are called as a negative space?
- The statement ‘’learnable prompts to facilitate the alignment  of visual and linguistic semantics at a shallow level but also utilizes explicit rationale to jointly encode multimodality by learning different prompts for each encoder layer.’’ should cite related papers.
- There are some minor errors in this paper. For example, “exploring to introduce programming approach” should be “exploring to introduce a programming approach”.

**Questions:**

See weakness.

**Limitations:**

See weakness.

---

> ### Author Rebuttal · Authors · 2023-08-10
>
> We thank the reviewer for the valuable feedback. Please find our responses below.
>
> > **General insights:** Are the analysis results of the single case in section 3.1.1 and 3.1.2 applicable to most samples in the data set?
> >
>
> Our findings in 3.1.1 and 3.1.2 are applicable to most samples. The applicability of “the different roles of rationales in zero-shot and fine-tuning learning” (3.1.1) possesses an intuitive nature and extends to generic scenarios. Similarly, the discovery “interleaved information without careful design exacerbates hallucinations” in section 3.1.2 remains pertinent for a substantial portion of the dataset, making it a noteworthy phenomenon. We present the quantitative analysis of hallucinations (see Appendix A.2 for details) and a new experiment for the discovery in 3.1.2 as depicted below:
>
> | Method | Authenticity  |
> | --- | --- |
> | w/o visual information | 0.883 |
> | w/ visual inforamtion | 0.602 |
> | w/ visual information & duty distinct w/o uncertainty  | 0.783 |
> | w/ visual information & duty distinct w uncertainty R | 0.855 |
>
> As shown in the table, introducing visual information exacerbates the hallucinations and diminishes the authenticity of generated rationales by 28.1% compared with naive prompting devoid of any visual information input. Our "duty distinct" and "uncertainty" designs effectively alleviate the hallucinations, elevating authenticity to a level comparable to scenarios without visual information input.
>
> Thanks for the valuable comments, and we will add the user study and analysis in the revised paper.
>
> > **Can gpt-3 reach similar conclusions as chatgpt?**
> >
>
> GPT-3 can reach similar conclusions as ChatGPT, in terms of ”difficulty in understanding dense image information” (3.1), ”rationale-sensitive reasoning” (3.1.1) and ”intensified hallucinations with visual information” (3.1.2). We provide additional illustrative instances in Appendix A.3, and the similar performance of our method when using GPT-3 and ChatGPT can also demonstrate that.
>
> > **On what basis are the three principles in 3.2 considered？**
> >
>
> We consider the three principles from the following insights:
>
> - For the principle “utilize LLMs’ intrinsic knowledge to generate multimodal rationales”: Rationales are necessary for multimodal comprehension with LLM because LLM has difficulty jointly reasoning multimodal information (3.1). Besides, flexible rationales are required to be knowledge-enriched (3.1.1), so LLMs become a natural choice for rationale generation.
> - For principle "explicitly cue the LLMs to differentiate the responsibilities of reasoning and recognition step by step": The challenge of hallucinations is exacerbated when introducing interleaved multimodal information as LLMs would fabricate visual information (3.1.2). Explicit duty distinct prompting is an optional solution for fabricated visual information, compelling the LLMs to identify the recognition responsibility of off-the-shelf visual models and visual models to obtain visual facts. Similarly, the responsibility for reasoning needs to be assigned to LLMs with reasoning capabilities.
> - For principle “explicitly mark the negative space for uncertain parts, emphasizing critical thinking in rationale generation”: Considering the rationale-sensitive reasoning for zero-shot prompting (3.1.1), the authenticity of rationales becomes significant. Uncertainty facilitates the LLMs to maintain a critical attitude towards questions and supplementary visual information, thereby enhancing the authenticity of the generated rationales.
>
> Thanks for the valuable feedback, and we will revise the overview of 3.2 to add the above clarification.
>
> > **Why the “uncertainty” are called as a negative space?**
> >
>
> The "negative space prompting" refers to our prompting method, including decomposition and uncertainty. The multi-modal CoT in our approach is decomposed into multiple sub-questions with "spaces", where the "space" that can be answered by the LLM is "positive", and otherwise the "space" is "negative" to be filled, i.e. the uncertainty. We intend to incorporate both decomposition and uncertainty into the name. If the clarification remains unclear, we will replace "negative space prompting" with "uncertainty prompting". Thank you for your feedback. We will carefully polish the paper and modify the confusing terms to facilitate understanding.
>
> > **Citation:** The statement ‘’learnable prompts to facilitate the alignment of visual and linguistic semantics at a shallow level but also utilizes explicit rationale to jointly encode multimodality by learning different prompts for each encoder layer.’’ should cite related papers.
> >
>
> Thanks for the valuable feedback, and we will cite and discuss the shallow prompting [1,2,3,4] and deep prompting [5,6,7,8] methods.
>
> > **Minor errors:** There are some minor errors in this paper. For example, “exploring to introduce programming approach” should be “exploring to introduce a programming approach”.
> >
>
> Thanks for pointing out this, and we will polish the paper in the revision.
>
> [1] Learning to prompt for vision-language models. Zhou, Kaiyang, et al. IJCV2022.
>
> [2] Conditional prompt learning for vision-language models. Zhou, Kaiyang, et al. CVPR2022.
>
> [3] Dualcoop: Fast adaptation to multi-label recognition with limited annotations. Sun, Ximeng, Ping Hu, and Kate Saenko. NeurIPS2022
>
> [4] Prompt distribution learning. Lu, Yuning, et al. CVPR2022.
>
> [5] Visual prompt tuning. Jia, Menglin, et al. ECCV2022.
>
> [6] Maple: Multi-modal prompt learning. Khattak, Muhammad Uzair, et al. CVPR2023.
>
> [7] P-tuning: Prompt tuning can be comparable to fine-tuning across scales and tasks. Liu, Xiao, et al. ACL2022.
>
> [8] P-tuning v2: Prompt tuning can be comparable to fine-tuning universally across scales and tasks. Liu X, Ji K, Fu Y, et al. arXiv:2110.07602, 2021.

---

### Official Review · Reviewer_kQGJ · 2023-07-04

**Soundness:** 3 good
**Presentation:** 3 good
**Contribution:** 2 fair
**Rating:** 6
**Confidence:** 4

**Summary:**

The work focuses on solving multimodal reasoning task. In the zero-shot setting, the work prompts LLM to conduct step-by-step reasoning. To avoid hallucination due to the lack of image features, LLM is asked to leave the answer to sub-questions as “uncertain” if they involve images. Then the corresponding sub-questions are answered by the visual components. In the fine-tuning setting, the generated rationales are used to train an LM augmented with a visual encoder. Experiments show that the elicited multimodal rationales can improve performance in both settings.

**Strengths:**

-	Originality: The work adopts careful prompt engineering that encourages LLM to (1) offload the step of visual recognition to the visual component and (2) discard errors in the intermediate results. Both are shown to be effective in mitigating hallucination and robust respectively.
-	Quality: The work investigates both zero-shot evaluation and fine-tuning to validate the effectiveness of the generated multimodal rationales. Comprehensive experiments and ablation studies are conducted to justify the design choices.
-	Clarity: The paper is well-written with illustrative running examples and figures. The intuitions behind each design choice are clearly stated, which helps readers to understand both the technical problems and motivations.
-	Significance: The proposed method brings significant improvement to a particular dataset, and also shows better generalization.


**Weaknesses:**

1.	The key ideas the paper tries to convey are somewhat scattered. The prompt engineering part and the visual component can be individually stand-alone as independent work. Right now, it is also hard to tell which part matters more. Perhaps the authors can add experiments where (1) a baseline model is trained on the rationales generated by the proposed method and (2) the proposed augmented model is trained on human-annotated rationales.
2.	Only one dataset is chosen. And many questions in ScienceQA do not require images to be answered indeed. I would encourage the authors to explore more suitable datasets.


**Questions:**

1.	How do you sample the rationales for fine-tuning? Do you only sample those leading to the correct predictions?

**Limitations:**

The authors have clearly stated the risks/biases brought by LLMs which are the common problems for LLMs.

---

> ### Author Rebuttal · Authors · 2023-08-10
>
> Thank you for your insightful and helpful comments. Our detailed responses are available below for your consideration.
>
> > **Somewhat scattered key ideas (part 1).** The prompt engineering part and the visual component can be individually stand-alone as independent work.
>
> We appreciate the review’s acknowledgment of our DDCoT prompting and visual components. Both DDCoT prompting and visual components facilitate inducing visual information to language models for multimodal reasoning. The former focuses on generating high-quality multimodal rationales. The latter aims at optimizing the utilization of those generated rationales and visual features, as visual information in images cannot be fully converted to the text inputs of LLMs. Thanks for your valuable feedback, and we will include a comprehensive perspective on the two technologies at the outset of Section 3.2.
>
> > **Somewhat scattered key ideas (part 2).** Which part of this paper matters more? Perhaps add experiments where (1) a baseline model is trained on the rationales generated by the proposed method and (2) the proposed augmented model is trained on human-annotated rationales.
>
> Thanks for the valuable suggestion. We also agree with the necessity to demonstrate the extent of impact exerted by the two components, and we present the results in the following table:
>
> |                   | IMG   | TXT   | Avg   |
> | ----------------- | ----- | ----- | ----- |
> | baseline          | 72.93 | 85.84 | 79.7  |
> | baseline + our R  | 75.81 | 82.40 | 82.83 |
> | baseline + gt R   | 81.07 | 84.07 | 85.97 |
> | our model         | 75.16 | 85.25 | 80.45 |
> | our model + our R | 83.34 | 91.2  | 87.34 |
> | our model + gt R  | 84.43 | 92.09 | 88.00 |
>
> As shown in the table, we observe that our R and visual components individually exhibit certain gains in terms of IMG improvement. However, **when combined, they yield substantial gains.** This observation aligns with our response above.
>
> Besides, please note that the annotated ground truth rationales within the ScienceQA dataset inherently encompass the final prediction, i.e., correct answers. To ensure a fair comparison, we manually exclude the answers from these annotations, using the remaining text as input rationales for fine-tuning. Under this configuration, our proposed rationale achieves a fine-tuning performance comparable to the annotated rationales.
>
> Thanks for pointing out this, and we will add the experiments and analysis in the revised paper.
>
> > **Only one dataset is chosen.** And many questions in ScienceQA do not require images to be answered indeed. I would encourage the authors to explore more suitable datasets.
>
> Thanks for your valuable feedback. The ScienceQA indeed includes some questions that do not require images to answer, but we achieve significant improvement in its IMG split (requiring images to answer). Currently, ScienceQA is a suitable dataset as it incorporates multimodality and needs for complex reasoning.
>
> We agree that extending our approach to more appropriate datasets would be beneficial. While other existing datasets may not fully exploit the benefits of our approach, we venture into exploring the captioning task on NoCaps and the video question-answering task on MSVD-QA:
>
> |      |  |  | NoCaps |                                                       | MSVD-QA |
> | ------ | ------ | ------------- | ------------------ | ------------------- | ------- |
> |        | CIDEr  | SkipThoughtCS | EmbeddingAverageCS | GreedyMatchingScore | Acc     |
> | BLIP-2 | 76.15  | 49.84         | 89.20              | 77.94               | 34.4    |
> | Ours   | 46.26  | 84.78         | 92.35              | 79.12               | 39.3    |
>
> **For captioning,** we prompt the LLM to solve sub-problems derived from a simple caption, aiming to optimize and enrich it using the corresponding sub-answers. The substantial knowledge within LLM enables the generation of semantically enriched captions, leading to improvements in metrics evaluating sentence semantics, i.e. 34.94%, 3.15%, and 1.18% in terms of SkipThoughtCS, EmbeddingAverageCS, and GreedyMatchingScore. Note that the CIDEr metric to evaluate ours is limiting. It is designed to measure the similarity between the tested caption and reference captions without considering the diversity and high-level semantics.
>
> **For video question answering,** LLM deconstructs problems like the decomposition step on ScienceQA. We sample video frames for VQA recognition and integrate frame information for multimodal rationale and answers. Leveraging the sequence understanding in LLM and visual information returned by the VQA model, we achieve a 4.9% improvement over BLIP-2.
>
> Due to time limitations, we randomly evaluated only 1000 images from NoCaps and 1000 videos from the MSVD test dataset in a zero-shot setting. In the revised paper, we intend to extend our experiments to encompass additional tasks in both zero-shot and fine-tuning settings.
>
> > **How do you sample the rationales for fine-tuning?** Do you only sample those leading to the correct predictions?
>
> For fine-tuning, we employ all the available rationales without resorting to sampling.

---

### Official Review · Reviewer_oVaJ · 2023-07-06

**Soundness:** 3 good
**Presentation:** 1 poor
**Contribution:** 2 fair
**Rating:** 6
**Confidence:** 3

**Summary:**

The paper proposes Duty-Distinct Chain-of-Thought (DDCoT) Prompting for multimodal reasoning problems (e.g. VQA). Despite CoT's success in language-only reasoning problems, authors argue that multimodal reasoning challenges CoT as the rationale part is sensitive to the input information. Since image caption is the only source of image information for LLM, once the caption is generated poorly, rationale will intensity hallucinations.

To this end, DDCoT proposes two insights: "keep critical thinking" and "disentangle reasoning and recognition". Specifically, the pipeline is like:
1. Given input question, zero-shot prompt LLM: “please think step-by-step and deconstruct the question down to necessary sub-questions"
2. Determine if each subquestion can be answered without visual information with zero-shot prompt: “Assume that you do not have any information about the picture, try to answer the sub-question and formulate the corresponding sub-answer as ‘Uncertain’ if the sub-question cannot be determined”. This helps deal with hallucination (i.e. LLM makes up facts about images).
3. Use an off-the-shelf VQA model to answer each subquestion "with negative space" (my guess is, each subquestion with 'Uncertain' as answer?)
4. Given subquestions and answers, LLM is prompted to perform CoT reasoning, generating a rationale and a final answer.

The paper also considers a finetuning setup with some architecture inventions.

Results on ScienceQA show
- zero-shot prompting: 1-3% improvement from previous few-shot CoT baselines
- finetuning: more impressive improvements and generalization from baselines.

Analysis on visual information source, rationale generation process, fine-tuning components, and explainability are also conducted to further justify DDCoT.


**Strengths:**

- Multimodal reasoning (e.g. VQA) is an important research direction, and LLM prompting is an emerging and promising approach to it. The paper highlights limitations of existing CoT-based methods for VQA (sensitive to caption; rationale does not improve performance), and propose reasonable solutions to these problems.

- DDCoT proposes an interesting heuristic way to combine LLM and off-the-shelf VQA models as a tool: LLM decomposes questions, decide which parts to call VQA model, and aggregates information back for answer. The approach could be potentially valuable for other tasks (involving video, code, webpages...)

- The consideration of both prompting and finetuning setups add to technical depth.

- Experiments show improvements, and their design looks solid. I appreciate comprehensive and diverse ablations and analysis.

**Weaknesses:**

- The presentation is a bit poor. The concrete problem is just "how to best put visual information in text, so that rationale can help", and the main insight is just "use VQA models across subquestions to best induce visual information, instead of off-the-shelf captioning". But it takes me some time to get it. I don't think "Flexibility/generalizability/explainability" makes too much sense to me --- I don't think few-shot CoT is really labor-intensive (just 3 examples??), inflexible, or hard to explain. The main motivation should be performance and generalization. 3.1.1 seems commonly known and a bit redundant. So intro, 3.1, 3.1.1, 3.1.2 are like four different stories of motivation, and a bit confusing to me. Why not sticking to one single story, instead of writing a story in 4 different ways/views/places?

- Continuing on the presentation issue, I don't get Figure 1,2,3,4,5 --- all are examples, and in fact, Figure 5 should better be the teaser given it actually tells a bit about how the method works. Fig 1's example doesn't really tell much, as MM-CoT and UnifiedQA's rationales are omitted. etc.

- Continuing on the presentation issue, some terms are confusing (or at least lack explaining), e.g. "negative space prompting".

- While finetuning performances seem stronger, prompting performance is only slightly better than few-shot CoT baselines, and there are not a lot of prompting baselines to begin with. In the long run, seems prompting performance will be more important than finetuning performance (one can also imagine multi-modal GPT-4 just solves ScienceQA and beats every model, making it less worthwhile to study...)

- The results are only on ScienceQA, not only how general the findings are, given original CoT is evaluated on various problems, and multimodal CoT should be able to solve a lot of different tasks as well.

**Questions:**

I think the paper can benefit from better writing, and experiments from different datasets.

**Limitations:**

Appendix E talks about hallucination and social bias.

---

> ### Author Rebuttal · Authors · 2023-08-10
>
> We thank the reviewer for the valuable feedback and the concerns about the presentation. We carefully address your questions and comments below and will improve the writing accordingly.
>
> > **The presentation is a bit poor (part 1)**: The concrete problem and main insight. The motivation and organization structure of Sec 3.
>
> Thanks for your detailed review. As the reviewer points out, our core problem is **how to best put visual information in the text to generate helpful rationales.** In order to address this problem, the below two explorations on helpful rationales and visual information are needed, which we present in Sec 3.1:
>
> - What kind of **rationales is helpful** and generalizable? (see Sec 3.1.1 for details)
> - What challenges and issues will there be in **putting visual information** for rationale generation? (see Sec 3.1.2 for details)
>
> Based on the explorations, we summarize key principles to generate helpful rationales and propose the concrete DDCoT prompting method in Sec 3.2:
>
> - As the reviewer points out, using VQA models across subquestions to best induce visual information is one of the main insights in DDCoT prompting.
> - In addition, the “sub-questions with negative-space prompting” (Line 191) and “integrate to joint reasoning” (Line 214) are also the main and crucial parts. The former is used to mitigate hallucination, and the latter help to generate coherent and logical rationale rather than scattered factual sub-questions and answers.
>
> The writing structure in “Section 3 Method” is presented from concept exploration (Sec 3.1) to concrete method design (Sec 3.2). We appreciate your valuable feedback, and we will provide an overview at the beginning of Section 3 to introduce our core problem, organizational structure, and main content of each subsection. Also, we will carefully revise the writing in each subsection accordingly.
>
> > **The presentation is a bit poor (part 2)**: I don't think few-shot CoT is really labor-intensive, inflexible, or hard to explain.
>
> It is labor-intensive for fully-supervised methods that require ground-truth rationale annotations. For few-shot CoT methods, we agree that providing a few examples of rationales for one case is not expensive. However, automatically searching for appropriate examples for a large set of diverse questions takes work (another promising direction, but not the scope of this paper), as CoT methods are greatly sensitive to the reasoning complexity and task-specific demonstration. That is also why we believe generalizability is crucial and currently focus on the zero-shot setting, which can generalize to different questions.
>
> > **Continuing on the presentation issue:** The information in Figures 1,2,3,4,5; Fig 5 should better be the teaser. Fig 1's example doesn't really tell much.
>
> As the reviewer raises in the previous weakness, “the main motivation should be performance and generalization,” we also agree that generalization and performance are very crucial aspects, which thus be presented in Fig 1 to be emphasized. Fig 1 (a) shows a simple out-of-distribution example to illustrate the poor generalization ability of previous methods compared to ours. And Fig 1 (b) shows the performance comparison in different settings. Figures 2, 3, and 4 are examples to help better demonstrate and understand our insights in Sec 3.1, 3.1.1, and 3.1.2, respectively.
>
> Thanks for the feedback and suggestion. We will combine our most crucial insights in Fig 4 and the method illustration in Fig 5 to redesign and update a more appropriate teaser.
>
> > **Continuing on the presentation issue:** Some terms are confusing (or at least lack explaining), e.g. "negative space prompting".
>
> The "negative space prompting" refers to our prompting method, including decomposition and uncertainty. We decompose multi-modal CoT into multiple sub-questions with "spaces", where the "space" are "positive" if LLMs can answer the sub-question, and otherwise, the "space" is "negative" to be filled, i.e. the uncertainty. We intend to include both decomposition and uncertainty in the name. If the explanation is still confusing, we will replace "negative space prompting" with "uncertainty prompting". Thank you for your feedback. We will carefully polish the paper and modify the confusing terms to facilitate understanding.
>
> > **Finetuning performances seem stronger.** Prompting performance will be more important.
>
> The performance improvement on finetuning is more significant than on zero-shot setting. This is understandable as the complete information of an image cannot be entirely translated into the text input for LLMs. Note that our prompting method with ChatGPT surpasses the few-shot CoT baseline by 4.61% in the IMG split (Line 285), which is significant and improves more than in the Avg split. Besides, we observe that our technique can benefit more when the used LLMs strengthen (2.53% for GPT-3 and 4.61% for ChatGPT) (Line 284).
>
> > **Multimodal CoT should be able to solve a lot of different tasks as well.**
>
> We appreciate the reviewer’s recognition of the potentially valuable of our approach, and we agree that multimodal CoT could be applied to various tasks as well. Per your advice, we further conduct experiments on captioning and video question answering tasks. The below table presents the general effectiveness of our approach across various tasks.
>
> |||| NoCaps|| MSVD-QA |
> | - | - | - | - | - | - |
> || CIDEr | SkipThoughtCS | EmbeddingAverageCS | GreedyMatchingScore | Acc|
> | BLIP-2 | 76.15 | 49.84| 89.20| 77.94| 34.4|
> | Ours   | 46.26 | 84.78| 92.35| 79.12| 39.3|
>
> Please refer to our third response to the Reviewer kQGJ for the analysis. Thanks for the valuable suggestion, and we will explore various tasks in zero-shot and finetuning settings in the revised paper.

---

> > ### Comment · Reviewer_oVaJ · 2023-08-16
> > **Thanks**
> >
> > increased my score from 5 to 6 in light of the rebuttal

---

> > > ### Author Response · Authors · 2023-08-16
> > > **Thank you for your encouraging response!**
> > >
> > > Thank you for your encouraging response! We deeply appreciate your thorough and invaluable feedback, and we will refine our paper accordingly.

---

### Official Review · Reviewer_ka68 · 2023-07-06

**Soundness:** 3 good
**Presentation:** 2 fair
**Contribution:** 3 good
**Rating:** 5
**Confidence:** 4

**Summary:**

1. The paper studied the challenges and limitations in rationale generation for multimodal problems. Then the authors propose a Duty-Distinct Chain-of-Thought Prompting (DDCoT) to collect language-related or visual-related information and select valid information to generate the rationale.
2. The rationale can be used in zero-shot and fine-tuning settings for question answering. The authors designed a fine-tuning framework with deep-layer prompting and rationale-compressed visual embedding.
3. The experiment results demonstrate the effectiveness of the rationale generated by the proposed method with SOTA results on the ScienceQA benchmark.

**Strengths:**

1. The proposed prompting design to separate the text reasoning and visual information for multi-modal QA problems is novel.
2. The result on the ScienceQA dataset is significant in zero-shot and fine-tuning settings, the ablation shows the effectiveness of different parts of the model.

**Weaknesses:**

1. Some experiment settings are not clearly explained or confused. See questions.
2. It's better to validate the effectiveness of 'Integrate to Joint Reasoning' part of prompting in the ablation if it's an important part of the method.

**Questions:**

1. The baseline(B) model in Table 2 is not explained.
2. Does 'B+our img' in Table 2 use rational-compressed visual embedding or not? What is the difference between 'B+our img' and 'w/ our R'? In line 308, it seems that the 'B+our img' includes rational-compressed visual embedding.
3. Does 'w/o DLP' in Table 3 include RCVE?

**Limitations:**

The limitation is not discussed in the paper. Societal impact is discussed.

---

> ### Author Rebuttal · Authors · 2023-08-10
>
> Thank you for the helpful review. We carefully address your questions and comments below.
>
> > **Some experiment settings** are not clearly explained or confused. See questions.
>
> Thank you for the valuable feedback, and we will clearly explain experimental settings in the revised paper and address the confusion. Below, we explain the questions one by one.
>
> > **The baseline(B) model in Table 2** is not explained.
>
> Following [1], we adopt the T5-Base with only text input (i.e. questions, contexts, and options) to predict answers as our baseline (B). All image-related information is omitted, given that T5 operates solely as a language model. Thanks for pointing out this, and we will include the explanation in the revision.
>
> [1] “Multimodal Chain-of-Thought Reasoning in Language Models” Zhang, Zhuosheng, et al. *arXiv preprint arXiv:2302.00923* (2023).
>
> > Does **'B+our img' in Table 2** use rational-compressed visual embedding or not? What is the difference between 'B+our img' and 'w/ our R'?
>
> - 'B+our img' in Table 2(b) uses both rational-compressed visual embedding and deep-layer prompting. The first four experiments in Table 2(b) focus on how to utilize visual information in different modalities. In comparison to captions, image modality presents a more challenging scenario for the model to comprehend visual information. Our proposed deep-layer prompting and rational-compressed visual embedding facilitate the model's understanding and learning of the alignment between different modalities, ultimately enabling the extraction and utilization of image features.
> - 'B+our img' in the first four lines is identical to 'no R' in the below three lines. The below three lines intend to elucidate the effects of rationales generated from different methods. Utilizing 'B+our img' as a foundation, the rationale produced by our DDCoT aids the model in better comprehending and reasoning for the multimodal context, resulting in notable performance improvement in the IMG split ('w/ our R').
>
> > Does **'w/o DLP' in Table 3** include RCVE?
>
> The 'w/o DLP' in Table 3 does not include RCVE. The notation 'w/o DLP' refers to the condition 'w/o DLP & w/o RCVE'. We will replace the notation 'w/o DLP' with 'w/o DLP & w/o RCVE' in our revised paper.
>
> > **Ablations on 'Integrate to Joint Reasoning':** It's better to validate the effectiveness of 'Integrate to Joint Reasoning' part of prompting in the ablation if it's an important part of the method.
>
> Thanks for the valuable feedback. We agree that validating the effectiveness of 'Integrate to Joint Reasoning' is essential, given its integral role within our approach.  We present the ablation studies for this component in the following table:
>
> |                  | IMG   | TXT   | Avg   |
> | ---------------- | ----- | ----- | ----- |
> | No R             | 75.16 | 85.25 | 80.45 |
> | naive R | 75.06 | 89.61 | 82.96 |
> | sub-qa as R      | 77.49 | 82.36 | 83.75 |
> | our R            | 83.34 | 91.2  | 87.34 |
>
> As shown in the table, the absence of the 'Integrate to Joint Reasoning' part also leads to a decline in performance, specifically -5.85% for the IMG split and -3.59% for overall performance. The decline is attributed to the model’s struggle to comprehend scattered facts instead of coherent reasoning chains, coupled with its limited reasoning capabilities.
>
> Thanks for the suggestion, and we will include the ablation and analysis of 'Integrate to Joint Reasoning' part of prompting in the revised paper.
>
> > **The limitation** is not discussed in the paper. Societal impact is discussed.
>
> We discuss the limitations across the following aspects (see Appendix. E for details): (1) The challenge of hallucinations in multimodal reasoning remains partially unresolved. (2) Cross-modality pretraining is anticipated to enhance the efficacy of our methods further. (3) The social biases introduced by the LLMs.

---

> > ### Comment · Reviewer_ka68 · 2023-08-18
> >
> > Thanks to the authors for the clarifications! However, the reviewer still has the following concerns:
> >
> > 1. Why only evaluate on ScienceQA benchmark? There are several visual reasoning that rationale generation may help such as A-OKVQA [1]. The authors propose a rationale-focus method. I believe it's better to discuss what kind of problems and datasets it suits.
> >
> > 2. The contribution is somewhat limited as the technical difference between RCVE module and former visual-text attention methods such as Q-former in BLIP2, perceiver in Flamingo, etc, seems limited [2,3,4]. Also, the novelty of DLP module, as mentioned by reviewer ndag.
> >
> > [1] A-OKVQA: A Benchmark for Visual Question Answering using World Knowledge
> > [2] BLIP-2: Bootstrapping Language-Image Pre-training with Frozen Image Encoders and Large Language Models
> > [3] Flamingo: a Visual Language Model for Few-Shot Learning
> > [4] Perceiver: General Perception with Iterative Attention

---

> > > ### Author Response · Authors · 2023-08-19
> > > **Response to concern #1: Benchmark**
> > >
> > > Thank you for the response and for posting the concerns to discuss!
> > >
> > > > Why only evaluate on ScienceQA benchmark? There are several visual reasoning that rationale generation may help such as A-OKVQA [1]. The authors propose a rationale-focus method. I believe it's better to discuss what kind of problems and datasets it suits.
> > > >
> > > - We evaluate on ScienceQA benchmark because (1) it is used to diagnose the **multi-hop reasoning ability and interpretability** when answering multimodal science questions. Such requirements of multi-hop reasoning and interpretability enable ScienceQA suitable for evaluating the ability in multimodal rationale generation (the rationale reveals **the interpretable process of multi-hop reasoning**) and the rationale’s effects in multimodal reasoning. (2) “the goal of SCIENCEQA is to aid development of a reliable model that is capable of generating **a coherent chain of thought** when arriving at the correct answer” (copy from ScienceQA paper). As we employ CoT prompting to generate rationales and use rationales as a kind of CoT to arrive at answers, ScienceQA is pretty suitable for evaluating our DDCoT prompting’s effectiveness. (3) ScienceQA features **rich** domain diversity (natural science, social science, and language science), context diversity, and level diversity with different grade-level science exams, and thus has multiple splits. These features are suitable for evaluating our method **in different and general situations**.
> > > - Although ScienceQA is the most suitest dataset, we agree with the reviewer that our rationale generation could be applied to other suitable datasets and tasks as well (reviewer oVaj and kQGJ also pointed out this). We have conducted experiments on NoCaps for the captioning task and MSVD-QA for the video question answering task, leading to performance improvement. Please see our responses to Reviewer oVaj or kQGJ for detailed results and analysis.

---

> > > ### Author Response · Authors · 2023-08-19
> > > **Response to concern #2: Technical Difference**
> > >
> > > > The contribution is somewhat limited as the technical difference between RCVE module and former visual-text attention methods such as Q-former in BLIP2, perceiver in Flamingo, etc, seems limited [2,3,4]. Also, the novelty of DLP module, as mentioned by reviewer ndag.
> > > >
> > >
> > > Thank you for your rigorous consideration. And we respond to our technical contributions and differences from former works in below three different aspects:
> > >
> > > - **Our contributions and technical novelties are far more than proposing modules (RCVE+DLP) for fine-tuning learning** (”Utilization for Fine-tuning Learning” in Sec 3.3). Instead, ”Utilization for Fine-tuning Learning” **is only one part of one of our three core contributions**, which is proposed to validate that our rationales are not only helpful for multimodal reasoning in zero-shot prompting but also in fine-tuning learning. Our core contributions and novelties are:
> > >     1. We are the **first** to study and achieve **zero-shot multimodal rationale generation**, considering flexibility, generalizability, and explainability. We deeply analyze the challenges of **putting visual information in the text to generate rationales** (Sec 3.1) and further **induct the critical principles** (Lines 187-190) in generating flexible and generalizable multimodal rationales using LLMs. These challenges (such as the discovery that LLMs’ hallucinations are intensified with interleaved multimodal inputs), analysis, and principles are not fully explored by previous works.
> > >     2. We propose zero-shot DDCoT prompting to generate multimodal rationales (Sec 3.2), which consists of three steps: (1) decompose the question into sub-questions **with negative-spacing prompting by considering the uncertainty.** Negative-space prompting with uncertainty brings 7.73% performance improvement. Please refer to Table 1 and Table 2 in Appendix, (2) visual recognition to obtain visual complements, and (3) **integration to joint reasoning** (see first-round “Rebuttal by Authors” for the importance of this design). The former methods do not explore these highlighted aspects.
> > >     3. We propose the utilization of rationales to improve LMs’ multimodal reasoning (Sec 3.3). To show the effectiveness and generalizability, we achieve multimodal Ireasoning in both settings: (1) **zero-shot prompting** and (2) **finetuning learning** (*the contribution mentioned by the reviewer*). Our methods significantly improve performance in both settings while exhibiting impressive generalization ability.
> > > - Regarding the finetuning modules themself, our RCVE and DLP are also different from former visual-text attention methods.
> > >     1. Q-former consists of two transformer submodules: image transformer and text transformer with share self-attention layers. The design is in order to make Q-former applicable to multiple different pre-training tasks simultaneously. In contrast, our approach utilizes a single transformer with cross-attention layers to compact visual embeddings from rationale input. And our finetuning modules are directly supervised by the training objectives of the downstream task.
> > >     2. Perceiver in Flamingo maps diverse-sized feature maps into a few visual tokens, disregarding the text context. In contrast, our RCVE first engages with the rationales generated by our DDCoT approach, and subsequently compresses visual features, taking into account the guidance provided by the rationales.
> > >     3. While DLP is a common design in prompt learning, it diverges in role in our work from previous research. Our DLP not only facilitates the alignment of vision and language modalities at a shallow level but also collaborates with RCVE to utilize rationales for joint multimodal encoding at every layer.
> > > - Our proposed rationale generation can be compatible with vision-language models, such as the mentioned Flamingo and recent Mini GPT-4. We first generate rationales by our DDCoT prompting and employ the rationales as extra inputs to Flamnigo and Mini GPT-4. Our DDCoT prompting significantly improves Flamingo and Mini GPT-4 by 11.14%, and 10.96%, respectively.
> > >
> > >
> > >     |  | NAT | SOC | LAN | TXT | IMG | NO | Avg |
> > >     | --- | --- | --- | --- | --- | --- | --- | --- |
> > >     | Flamingo | 21.89 | 52.41 | 20.27 | 23.50 | 39.11 | 19.02 | 27.87 |
> > >     | Flamingo with our rationales | 39.20 | 48.93 | 30.90 | 39.68 | 45.81 | 32.40 | 39.01 |
> > >     | Mini GPT-4 | 43.83 | 48.59 | 43.36 | 55.01 | 42.84 | 41.67 | 44.71 |
> > >     | Mini GPT-4 with our rationales | 57.37 | 62.32 | 46.82 | 65.91 | 56.72 | 48.57 | 55.67 |

---

### Author Rebuttal · Authors · 2023-08-10

We really appreciate all reviewers for their valuable feedback. Our code will be made public upon acceptance.

We are encouraged by the reviewers’ recognition of our novel/interesting contribution (ka68, oVaJ, aDZq), solid and robust technical design (oVaJ, kQGJ, aDZq), compelling performance improvement (ka68, oVaJ, kQGJ, aDZq, ndag) and diverse ablation studies (ka68, oVaJ, kQGJ).

------

In our individual replies, we attempted to address specific questions and comments as clearly and detailed as possible. Here, we briefly summarize these additional experiments and evaluations:

- Ablations on the 'Integrate to Joint Reasoning' part.
- Additional experiments on Captioning and Video Question Answering tasks.
- Quantitative ablations on our DDCoT and visual components.
- Additional user study for validating the generalization of the discovery in 3.1.2.
- Quantitative comparison with existing pre-trained VLMs and multimodal reasoning models: Our proposed rationale generation is compatible with such pre-trained VLMs.

------

We hope that these additional results further strengthen DDCoT’s position as the state-of-the-art multimodal Chain-of-Thought (CoT) approach:

- We are the **FIRST** to study and achieve **zero-shot multimodal rationale generation**, considering flexibility, generalizability, and explainability.
- We deeply analyze the challenges of **putting visual information in the text to generate rationales** (Sec 3.1) and **induct the critical principles** (Lines 187-190) in generating flexible and generalizable multimodal rationales using LLMs: (1) Drawing on the LLMs' intrinsic knowledge, (2) differentiating the responsibilities of reasoning and recognition, and (3) emphasizing critical thinking in the face of uncertainty.
- We propose zero-shot DDCoT prompting to generate multimodal rationales that significantly improve the multimodal reasoning abilities of LMs in both zero-shot prompting and fine-tuning learning while exhibiting impressive generalization ability.

---

### Decision · Program_Chairs · 2023-09-21

**Decision:**

Accept (poster)

**Comment:**

The paper proposes a novel method to generate multimodal rationales for question answering using language models and visual models. The method improves the performance and robustness of the system and reduces the hallucination problem. The paper also conducts extensive experiments and analysis to validate the effectiveness of the method. Some questions were raised by reviewers and the authors have already answered them in the rebuttal phase, such as the definitions of some experimental settings, the differences between this work and previous work, the claim of the key contribution (i.e., the 1st to study and achieve zero-shot multimodal rationale generation), the results of the proposed method on more datasets, etc. Based on the consensus got from all reviewers, I recommend an ACCEPT to this paper.